# Towards optimally abstaining from prediction with OOD test examples

**Adam Tauman Kalai**
Microsoft Research

**Varun Kanade**
University of Oxford

## Abstract

A common challenge across all areas of machine learning is that training data is not distributed like test data, due to natural shifts, "blind spots," or adversarial examples; such test examples are referred to as *out-of-distribution* (OOD) test examples. We consider a model where one may abstain from predicting, at a fixed cost. In particular, our *transductive* abstention algorithm takes labeled training examples and unlabeled test examples as input, and provides predictions with optimal prediction loss guarantees. The loss bounds match standard generalization bounds when test examples are i.i.d. from the training distribution, but add an additional term that is the cost of abstaining times the statistical distance between the train and test distribution (or the fraction of adversarial examples). For linear regression, we give a polynomial-time algorithm based on Celis-Dennis-Tapia optimization algorithms. For binary classification, we show how to efficiently implement it using a proper agnostic learner (i.e., an Empirical Risk Minimizer) for the class of interest. Our work builds on a recent abstention algorithm of Goldwasser, Kalais, and Montasser [10] for transductive binary classification.

## 1 Introduction

For learning of a class of functions $F$ of bounded complexity, statistical learning theory guarantees low error if test examples are distributed like training examples. Thus abstention is not necessary for standard realizable prediction. However, when the test distribution $Q$ is not the same as the distribution of training examples $P$, abstaining from prediction may be beneficial if the cost of abstaining $\alpha$ is substantially less than the cost of an error. This is particularly important when there are "blind spots" where $Q(x) > P(x) = 0$. Such blind spots may occur because of natural distribution shifts, or indeed may be due to adversarial attacks. Collectively, test examples that deviate from the training distribution are referred to as *out of distribution* (OOD) test examples. Such an extreme *covariate shift* scenario was analyzed for binary classification in recent work by Goldwasser, Kalais, and Montasser [10] (henceforth GKKM).

Abstaining from predicting may be useful because it is well-known to be impossible to guarantee accuracy on test examples from arbitrary $Q \neq P$ (abstention is unnecessary under the common assumption that $Q(x)/P(x)$ is upper-bounded) [7]. Whether one is classifying images or predicting the probability of success of a medical procedure, training data may miss important regions of test examples. As an example, Fang et al. [9] observed noticeable signs of COVID-19 in many lung scans; any model trained on data pre-2019 would not have any such instances in its training dataset. In the case of image classification, a classifier may be trained on publicly available data but may be also used on people's private images or even adversarial spam/phishing images [21]. In settings such as medical diagnoses or content moderation, abstention may be much less costly than a misclassification. In practice, if a model abstains from making a prediction, it could be followed up by a more elaborate (and costly) model or direct human intervention. The extra costs of human intervention, or as may be the case in a medical setting, a more expensive test, needs to be traded off with the potential costs of

misclassification. For a regression example, a model may be used to predict the success probability of a medical treatment across a population, yet examples from people that are considered high-risk for the treatment may be absent from the training data.

We begin by discussing binary classification, and then move to regression. In particular, for any distribution $P$ over examples $x \in X$ and any true classifier $f : X \to \{0, 1\}$ in $F$ of VC-dimension $d$, the so-called Fundamental Theorem of Statistical Learning (FTSL) guarantees w.h.p. $\tilde{O}(d/n)$ error rate[1] on future examples from $P$ using any classifier $h \in F$ that agrees with $f$ on $n$ noiseless labeled examples [see, e.g., 20].

In Chow's original abstention model [6], a *selective classifier* is allowed to either make a prediction $\hat{y}$ at a loss of $\ell(y, \hat{y}) \geq 0$ or abstain from predicting at a fixed small loss $\alpha > 0$. Following GKKM, we consider a *transductive* abstention algorithm that takes as input $n$ unlabeled test examples and $n$ labeled training examples and predicts on a subset of the $n$ test labels, abstaining on the rest. The goal is to minimize the average loss on the test set. (Unfortunately, the natural idea to abstain on test points whose labels are not uniquely determined by the training data can lead to abstaining on all test points even when $P = Q$.) GKKM give a algorithm with guarantees that naturally extend the FTSL to $Q \neq P$ albeit at an additional cost. The word transductive refers to the prediction model where one wishes to classify a given test set rather a standard classifier that generalizes to future examples, though the two models are in fact equivalent in terms of expected error as we discuss. The term *covariate shift* is appropriate here as it describes settings in which $Q \neq P$ but both train and test labels are consistent with the same $f \in F$. Without abstention, both transductive learning and covariate shift have been extensively studied [see,e.g., 1].

The principle behind our approach is illustrated through an example of Figure 1. Suppose $P$ is the distribution $Q$ restricted to a set $S \subset X$ which contains, say, 90% of the test set, so that 10% of the unlabeled test examples are in blind spots. Say we have learned a standard classifier $h \in F$ from the $n$ labeled training examples. Hypothetically, if we knew $S$, then we could predict $h(x_i)$ for test $x_i \in S$ and abstain from predicting on $x_i \notin S$. If abstaining costs $\alpha$, then the FTSL would naturally guarantee test loss $\leq 0.1\alpha + \tilde{O}(d/n)$, because one abstains on 10% of the test examples and the remaining test examples are distributed just like $P$. The difficulty is that $S$ may be too complex to learn. To circumvent this, we suggest a conceptually simple but theoretically powerful approach: choose the set of points to abstain on so as to minimize the worst-case loss over any true function $f \in F$ that is also consistent with the training data. In particular, given a predictor $h$ and a set of test points not abstained on, (ignoring efficiency) one could compute the worst-case loss over all $f \in F$ that are consistent with the labeled training examples. This approach achieves optimal worst-case guarantees and, in particular inherits the natural loss guarantee one attains from abstaining outside of $S$ discussed above. Converting this theoretical insight into an efficient algorithm is the focus of this paper, and different algorithms are needed for the case of classification and regression.

**Interpretation.** In the case of known $P, Q$, one may say there is a *known unknown*: the region of large $Q(x)/P(x)$. In our model, however, even though this region is an *unknown unknown*, we achieve essentially the same bounds as if we knew $P$ and $Q$. Thus, perhaps surprisingly, there is little additional loss for not knowing $P$ and $Q$. GKKM give related guarantees, but which suffer from not knowing $P, Q$. In particular, even when $P = Q$ their guarantees are $\tilde{O}(\sqrt{d/n})$ compared to the $\tilde{O}(d/n)$ of FTSL, and they give a lower-bound showing $\Omega(\sqrt{d/n})$ is inherent in their "PQ" learning model. In the PQ model, the error rate on $Q$ (mistakes that are not abstained on) and the abstention rate *on future examples from $P$* are separately bounded. In Appendix C, we show that the Chow model is stronger than PQ-learning in the sense that our guarantees imply PQ-bounds similar to theirs, but the reverse does not hold. Hence, the algorithms and guarantees in this paper extend the FTSL in both the Chow and PQ models. We also note that abstaining on predictions can either help reduce inaccuracies on marginal groups or be a source of unfairness towards a person or group of people, and it should not serve as an excuse not to collect representative data.

## 1.1 Classification results

For classification, suppose $Y = \{0, 1\}$, $\ell(y, \hat{y}) := |y - \hat{y}|$ is the 0-1 loss (more general losses are considered in the body), and $F$ is a family of functions of VC-dimension $d$. There is also a deterministic

---

[1]The $\tilde{O}$ notation hides logarithmic factors.

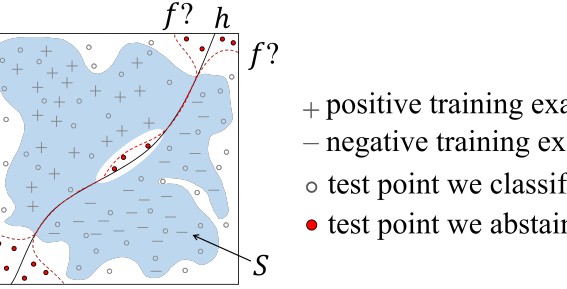

+ positive training example
− negative training example
∘ test point we classify
• test point we abstain on

Figure 1: A simple illustration of our approach when the training distribution $P$ is the test distribution $Q$ restricted to a (blue) set $S$. We use classifier $h$ fit on the labeled training data. We choose the subset of test examples to abstain on so as to minimize the worst-case loss over possible $f$'s consistent with the training data. This gives a better worst-case loss than if we knew $S$ and did the obvious thing of abstaining on all test $x \notin S$.

Empirical Risk Minimization (ERM) oracle that computes $\mathsf{ERM}(\mathbf{x}, \mathbf{y}) \in \arg\min_{g \in F} \sum_i \ell(y_i, g(x_i))$ on any dataset $\mathbf{x} \in X^n, \mathbf{y} \in Y^n$ (even noisy). While it is NP-hard to efficiently compute ERM for many simple classes like disjunctions, previous reductions to ERM have proven useful with off-the-shelf classifiers (e.g., neural networks) albeit without theoretical guarantees.

The inputs to the learner are labeled training examples $\bar{\mathbf{x}} \in X^n, \bar{\mathbf{y}} = f(\bar{\mathbf{x}}) \in Y^n$, and unlabeled test examples $\mathbf{x} \in X^n$. For transductive learning, the goal is to predict labels for the $n$ test examples $\mathbf{x}$. A transductive abstention algorithm is also given the predictor $h := \mathsf{ERM}(\bar{\mathbf{x}}, \bar{\mathbf{y}})$, which has 0 training error, and selects a vector $\mathbf{a} \in [0,1]^n$ for the probability of abstaining on test examples $\mathbf{x}$. Its loss is defined to be,

$$\ell_{\mathbf{x}}(f, h, \mathbf{a}) := \frac{1}{n} \sum_{i=1}^{n} a_i\, \alpha + (1 - a_i)\ell(f(x_i), h(x_i)). \tag{1}$$

Our Min-Max Abstention (MMA) reduction, mentioned above, attempts to minimize the maximum test loss among classifiers consistent with the training data. In particular, it (approximately) solves the following convex optimization problem for $\mathbf{a}$:

$$\min_{\mathbf{a} \in [0,1]^n} \max_{g \in V} \ell_{\mathbf{x}}(g, h, \mathbf{a}). \tag{2}$$

Here, $V := \{g \in F : \forall i\; g(\bar{x}_i) = f(\bar{x}_i)\}$ is the *version space* of classifiers consistent with the training labels; thus $h, f \in V$. In other words, MMA minimizes the worst-case test loss it could possibly incur based on the labeled training and unlabeled test data. A key insight is that (2) has *no unknowns*, so MMA will achieve a max in (2) as low as if it knew $P, Q$ and even $f$.

To solve (2), one must be able to *maximize* loss over $V$. Fortunately, GKKM showed how to solve that using a simple subroutine, which we call FLIP, that calls ERM.

**Theorem 1** (Classification). *For $Y = \{0, 1\}$, any $n, d \in \mathbb{N}$, any $F$ of VC dimension $d$, any $f \in F$, and any distributions $P, Q$ over $X$,*

$$\mathbb{E}_{\bar{\mathbf{x}} \sim P^n, \mathbf{x} \sim Q^n} [\ell_{\mathbf{x}}(f, h, \hat{\mathbf{a}})] \leq \alpha |P - Q|_{TV} + \frac{2d \lg 3n}{n},$$

*where $h = \mathsf{ERM}(\bar{\mathbf{x}}, f(\bar{\mathbf{x}}))$ and $\hat{\mathbf{a}} = \mathsf{MMA}(\bar{\mathbf{x}}, f(\bar{\mathbf{x}}), \mathbf{x}, h, \mathsf{FLIP}) \in [0,1]^n$ can be computed in time* $\mathrm{poly}(n)$ *using the* ERM *oracle for $F$.*

The total variation distance $|P - Q|_{\mathrm{TV}}$, also called the statistical distance, is a natural measure of non-overlap that ranges from 0 when $P = Q$ to 1 when $P$ and $Q$ have disjoint supports. The $\alpha |P - Q|_{\mathrm{TV}}$ term arises because the learner may need to abstain where $P$ and $Q$ do not overlap, and the other term derives directly from the same bound one gets in the case where $P = Q$ (so $|P - Q|_{\mathrm{TV}} = 0$). These bounds are stated in terms of expected loss, because unfortunately as we show in Lemma 8, high probability bounds require $\Omega(\alpha/\sqrt{n})$ loss because of the variance in how $Q$ samples are distributed. All proofs, unless otherwise stated, are deferred to Appendix F.

We generalize Theorem 1 in multiple ways. First, we point out a stronger bound in terms of a divergence $D_k(P\|Q)$, $k \geq 1$, that measures the excess of $Q(x)$ over $k \cdot P(x)$. It generalizes the total variation distance $|P - Q|_{\mathrm{TV}}$ between distributions $P, Q$:

$$D_k(P\|Q) := \sum_x \max\big(Q(x) - k \cdot P(x), 0\big) \in [0, 1] \tag{3}$$

$$|P - Q|_{\mathrm{TV}} := D_1(P\|Q) = \frac{1}{2}\sum_x |P(x) - Q(x)| \tag{4}$$

Note that if $Q(x) \leq k \cdot P(x)$ for all $x$, then $D_k(P\|Q) = 0$.

Second, we show this implies generalization by using a transductive abstaining algorithm to bound the expected loss with respect to future examples from $Q$. That is, to go alongside classifier $h : X \to \{0, 1\}$, we can output *abstainer* $\mathcal{A} : X \to [0, 1]$ that gives a probability of abstaining on each test example. Here, the generalization loss is,

$$\ell_Q(f, h, \mathcal{A}) := \mathop{\mathbb{E}}_{x \sim Q}\big[\alpha\,\mathcal{A}(x) + (1 - \mathcal{A}(x))\ell\big(f(x), h(x)\big)\big].$$

These two generalizations are summarized by the following theorem, which we state for classification but also has a regression analog.

**Theorem 2** (Generalization for classification)**.** *Fix $Y = \{0, 1\}$ and $F$ of VC dimension $d$. For any $f \in F$, and any distributions $P, Q$ over $X$,*

$$\mathop{\mathbb{E}}_{\bar{\mathbf{x}} \sim P^n, \mathbf{x} \sim Q^n}[\ell_Q(f, h, \mathcal{A})] \leq \min_{k \geq 1} \alpha D_k(P\|Q) + \frac{2dk \lg 3n}{n} \leq \alpha|P - Q|_{TV} + \frac{2d \lg 3n}{n}.$$

*where $h = \mathsf{ERM}(\bar{\mathbf{x}}, f(\bar{\mathbf{x}}))$ and abstainer $\mathcal{A} : X \to [0, 1]$ can be computed in time $\mathrm{poly}(n)$ using* ERM *and is defined by $\mathcal{A}(x') := \mathsf{MMA}\big(\bar{x}, f(\bar{x}), (x', x_2, \ldots, x_n), h, \mathsf{FLIP}\big)$.*

These guarantees use the same algorithm—to predict on a new test example $x'$, it simply runs MMA with a modified test set where we have replaced the first test example by $x'$ and returns the probability of abstaining on it.

Third, as in GKKM, guarantees hold with respect to a "white-box" adversarial model in which there is only a training distribution $P$ and an adversary who may corrupt any number of test examples (but they are still labeled by $f$). More specifically, natural train and test sets $\bar{\mathbf{x}}, \mathbf{z} \sim P^n$ are drawn, and an adversary may form an arbitrary test set $\mathbf{x} \in X^n$. We achieve guarantees as low as if one knew exactly which examples were corrupted and abstained on those:

**Theorem 3** (Adversarial classification)**.** *For any $f \in F$ with $d = \mathsf{VC}(F)$, any $n \in \mathbb{N}, \delta \geq 0$ and any distribution $P$ over $X$, with probability $\geq 1 - \delta$ over $\bar{\mathbf{x}}, \mathbf{z} \sim P^n$, the following holds simultaneously for all $\mathbf{x} \in X^n$:*

$$\ell_{\mathbf{x}}(f, h, \hat{\mathbf{a}}) \leq \frac{\alpha}{n}\,|\{i : x_i \neq z_i\}| + \frac{2d \lg 2n + \lg 1/\delta}{n}.$$

*where $h = \mathsf{ERM}(\bar{\mathbf{x}}, f(\bar{\mathbf{x}}))$ and $\hat{\mathbf{a}} = \mathsf{MMA}(\bar{\mathbf{x}}, f(\bar{\mathbf{x}}), \mathbf{x}, h, \mathsf{FLIP}) \in [0, 1]^n$ can be computed in time* $\mathrm{poly}(n)$ *using* ERM.

In their adversarial setting, GKKM again has $\tilde{O}(\sqrt{d/n})$ bounds. Our presentation actually begins in this adversarial framework as it is in some sense most general–adversarial robustness implies robustness to covariate shift.

## 1.2 Linear regression results

For selective linear regression, the classic problem with transductive abstention, we give a more involved but fully polynomial-time algorithm. (GKKM did not discuss regression.) Here, $Y = [-1, 1]$, $X = B_d(1)$ is taken to be the unit ball in $d$ dimensions. There is now a joint distribution $\nu$ over $X \times Y$ such that: $f(x) := \mathop{\mathbb{E}}_{\nu}[y|x] = w \cdot x$ for some vector $w$ with $\|w\| \leq 1$. We write $(\bar{\mathbf{x}}, \bar{\mathbf{y}}) \sim \nu^n$ to indicate that the $n$ labeled training examples are drawn from $\nu$. The loss for selective regression $\ell_{\mathbf{x}}(h, f, \mathbf{a})$ is still defined as in (1) except that $\ell(f(x), h(x)) := |f(x) - h(x)|^2$. Note that we are considering loss with respect to $f$ rather than $y$ which means that it may approach 0 for identical train and test distributions. Indeed, the additional loss term we will face due to the covariate shift is again $\alpha|P - Q|_{\mathrm{TV}}$, where $P$ and $Q$ are the marginal distributions over $X$ for $\nu$ and the test distribution, respectively.

**Theorem 4** (Linear regression)**.** *Let* $Y = [-1, 1]$*,* $n, d \in \mathbb{N}$*,* $\delta > 0$*, and* $X = B_d(1)$*. Let* $P, Q$ *be distributions over* $X$ *and* $\nu$ *be a distribution over* $X \times Y$ *with marginal* $P$ *over* $X$*. Let* $f(x) := \underset{(x,y)\sim\nu}{\mathbb{E}} [y|x] = w \cdot x$ *for some* $w \in B_d(1)$*. Then,*

$$\underset{(\bar{\mathbf{x}},\bar{\mathbf{y}})\sim\nu^n, \mathbf{x}\sim Q^n}{\mathbb{E}} [\ell_{\mathbf{x}}(f, h, \hat{\mathbf{a}})] \leq \alpha |P - Q|_{\mathrm{TV}} + \frac{\kappa \log n}{\sqrt{n}}.$$

*where* $\kappa$ *is a constant,* $h = \mathsf{ERM}(\bar{\mathbf{x}}, \bar{\mathbf{y}})$ *and* $\hat{\mathbf{a}} = \mathsf{MMA}(\bar{\mathbf{x}}, \bar{\mathbf{y}}, \mathbf{x}, h, \mathsf{CDT}) \in [0, 1]^n$ *can be computed in time* $\mathrm{poly}(n, d)$*.*

A technical challenge is that, as a subroutine, we need to find a linear model maximizing squared error, which is a non-convex problem. Fortunately, this step can be formulated as a Celis-Dennis-Tapia (CDT) problem, for which efficient algorithms are known [5, 3]. Our information theoretic results generalize to more general regression in a straightforward fashion, however the algorithm for maximizing loss is non-convex and we only know how to implement it efficiently for linear regression.

**Contributions and organization.** The main contributions of this work are: (1) introducing and analyzing an approach for optimally abstaining in classification and regression in the transductive setting, and (2) giving an efficient abstention algorithm for linear regression and an efficient reduction for binary classification (optimal in our model). After reviewing related work, we formulate the problem in a manner that applies to both types of prediction: binary classification and regression. Section 2 gives information-theoretic bounds for selective prediction. Section 3 covers the general MMA reduction. We then focus on classification, where Section 4 gives bounds and an efficient reduction to ERM. Finally, Section 5 gives bounds and a polynomial-time abstention algorithm for linear regression.

## 1.3   Related work

There is relatively little theoretical work on selective classification tolerant to distributional shift. Early work had algorithms that could learn certain binary classes $F$ with very strong absolute abstention and error guarantees [17, 13, 18]. However, it was impossible to learn other simple concept classes, even rectangles, in their models. GKKM was the first work to provide abstention guarantees for arbitrary classes of bounded VC dimension, achieving this with bounds that depended on $|P - Q|_{\mathrm{TV}}$. They showed that unlabeled examples (i.e., transductive learning) were provably necessary. In their PQ-learning model, specific to binary classification, they simultaneously guarantee $\tilde{O}(\sqrt{d/n})$ errors ($h(x) \neq f(x)$ which are *not* abstentions) from distribution $Q$ and $\leq \epsilon$ rejections *on future examples from distribution* $P$. While this latter condition may seem counter-intuitive at first, it implies guarantees in terms of total variation distance similar to the ones we give. However, they show that this approach to deriving bounds necessarily suffers from a $\Omega(\sqrt{d/n})$ inherent loss. We circumvent this lower-bound in the Chow cost model by directly optimizing over $Q$ without regard to rejections on $P$. Appendix C details this comparison.

Other related work is either about selective classification or distributional shift, but not both. Work on selective classification, also called "reliable learning" and "classification with a reject option," [6, 2, 8] study the benefits of abstaining in a model with fixed small cost $\alpha > 0$. The goal is to abstain in regions where the model is inaccurate. Recent work by Bousquet and Zhivotovskiy [4] gives a (not necessarily computationally efficient) algorithm for obtaining fast rates with the *Chow loss* whenever the cost of abstention is bounded away from $1/2$. They use this to characterize when fast rates for classification may be possible in the agnostic setting with improper learning. Finally, among the large body of prior work on transductive learning, the closest is work on transfer learning (without abstentions) [1]. Prior work on distributional shift without abstention [see, e.g., the book 16] has developed refined notions of distributional discrepancy with respect to classes of binary or real-valued functions [14]. Since the focus of our work is on abstention, we use total-variation distance. In future work, it would be interesting to see if these notions could be adapted to our setting.

Note that while our work addresses adversarial examples, it is very different from the recent line of work focused on robustness to small perturbations of the input [19, 11]. While our algorithms are not robust to such perturbations, they may at least abstain from predicting on such examples rather than misclassifying them.

## 1.4 Definitions

A table of our notation is given in Appendix A for convenience. Many of the definitions are common to classification and regression. Let $X$ denote the instance space[2] and $Y$ denote the label space, and let $F$ be a known family of functions from $X$ to $Y$. There is an unknown target function $f \in F$ to be learned.

We use boldface to indicate vectors (indexing examples), e.g., $\mathbf{a} = (a_1, \ldots, a_k)$. We denote by $f(\mathbf{a}) = (f(a_1), \ldots, f(a_k))$ for function $f$. Throughout the paper, we assume that $\alpha$ and $F$ are fixed and known. Let $[n] = \{1, 2, \ldots, n\}$ and $\mathbf{1}[\phi]$ denotes the indicator that is 1 if predicate $\phi$ holds and 0 otherwise.

There is a general base loss $\ell : Y \times Y \to \mathbb{R}_+$. We extend the definition of loss to loss on a sequence $\mathbf{x}$ and with abstention indices $A \subseteq [n]$:

$$\ell_{\mathbf{x}}(f, h) := \frac{1}{n} \sum_{i=1}^{n} \ell(f(x_i), h(x_i)), \quad \ell_{\mathbf{x}}(f, h, A) := \frac{\alpha}{n}|A| + \frac{1}{n} \sum_{i \notin A} \ell(f(x_i), h(x_i)).$$

We assume access to a deterministic algorithm returning $\mathsf{ERM}(\mathbf{x}, \mathbf{y}) \in \arg\min_{g \in F} \sum_i \ell(y_i, g(x_i))$ that runs in unit time on any dataset $\mathbf{x} \in X^n, \mathbf{y} \in Y^n$ (even noisy).

The algorithms all take labeled training examples $\bar{\mathbf{x}} \in X^n, \bar{\mathbf{y}} \in Y^n$ and unlabeled test examples $\mathbf{x} \in X^n$ as input. For binary classification, as discussed, $Y = \{0, 1\}$, there are train and test distributions $P, Q$ over $X$, and all examples are assumed to be labeled by $f$, so $\bar{\mathbf{y}} = f(\bar{\mathbf{x}})$. Here, the version space of classifiers with 0 training error is defined to be $\mathrm{VS}(\bar{\mathbf{x}}, \bar{\mathbf{y}}) := \{g \in F : g(\bar{\mathbf{x}}) = \bar{\mathbf{y}}\}$.

For regression, $Y = [-1, 1]$ and there is a train distribution $\nu$ over $X \times Y$ with marginal $P$ over $X$, and it is assumed that $f(\bar{x}) := \mathbb{E}_{(\bar{x}, \bar{y}) \sim \nu}[\bar{y}|\bar{x}] \in F$. The test distribution similarly has marginal $Q$ over $X$. The version space is $\mathrm{VS}_\epsilon$ is somewhat more complicated and is defined in Section 5.

With respect to $V \subseteq F$ (which we will generally take to be the version space), we denote the worst-case test loss of a given classifier $h$ (with a given abstention set $A$) on the test set, denoted by $\mathrm{L}_{\mathbf{x}}$:

$$\mathrm{L}_{\mathbf{x}}(V, h) := \max_{g \in V} \ell_{\mathbf{x}}(g, h), \quad \mathrm{L}_{\mathbf{x}}(V, h, A) := \max_{g \in V} \ell_{\mathbf{x}}(g, h, A). \tag{5}$$

## 2 Information-theoretic bounds for classification and regression

This section bounds the statistical loss (ignoring computation time) of choosing where to abstain $A$ so as to minimize the upper-bound $\mathrm{L}_{\mathbf{z}}(V, h, A)$ on test loss, where the predictor $h$ (used when not abstaining) is first fit on training data and $V \subseteq F$ is a general version space containing the target $f$. For noiseless binary classification, $V$ would simply be those classifiers consistent with $f$, and $V$ is a bit different for regression. Now, for "normal" test data $\mathbf{z} \sim P^n$ iid from the train distribution, a standard generalization bound would apply to the worst-case test loss $\mathbb{E}_{\mathbf{z} \sim P^n}[L_{\mathbf{z}}(V, h)]$ (e.g., $\tilde{O}(d/n)$ for binary classification with 0-1 loss). Our bounds are stated in terms of this quantity $\mathbb{E}_{\mathbf{z} \sim P^n}[L_{\mathbf{z}}(V, h)]$ and can be viewed as the overhead due to $P \neq Q$.

### 2.1 Learning with adversarial test examples

We begin with the adversarial setting of GKKM since it is particularly simple and will imply bounds in a distributional case. Here, there is only one distribution $P$. First, nature picks $n$ "natural" iid train and test examples $\bar{\mathbf{x}}, \mathbf{z} \sim P^n$. As mentioned, if there were no adversary, a learning algorithm might output some $h$ in a version space $V$ and would have a test error guarantee of $\mathrm{L}_{\mathbf{z}}(V, h)$.

Instead, a computationally-unlimited adversary armed with knowledge of $\bar{\mathbf{x}}, \bar{\mathbf{y}}, \mathbf{z}, f$, and the learning algorithm (and its random bits), chooses an arbitrary test set $\mathbf{x} \in X^n$ by corrupting as many or few test examples as desired. *If we knew* $\mathbf{z}$, the adversary could corrupt any $\gamma$ fraction of examples, we could of course guarantee loss $\leq \gamma\alpha + \mathrm{L}_{\mathbf{z}}(V, h)$ by abstaining on the modified examples where $x_i \neq z_i$. While the learning algorithm does not see $\mathbf{z}$, it can still achieve the same guarantee:

---

[2]To avoid measure-theoretic issues, for simplicity we assume that $X$ is finite or countably infinite.

**Lemma 1.** *[Adversarial loss] For any* $n \in \mathbb{N}, V \subseteq F, f \in V, \mathbf{z}, \mathbf{x} \in X^n, h : X \to Y$ *and all* $A^* \in \arg\min_{A \subseteq [n]} \mathrm{L}_{\mathbf{x}}(V, h, A)$:

$$\ell_{\mathbf{x}}(f, h, A^*) \leq \frac{\alpha}{n} \big| \{i : x_i \neq z_i\} \big| + \mathrm{L}_{\mathbf{z}}(V, h).$$

What this means is that if the natural training and test distributions are iid from the same distribution, then no matter which examples an adversary chooses to corrupt—$\mathbf{x}$ may be arbitrary—if one chooses $A$ to minimize $\mathrm{L}_{\mathbf{x}}(V, h, A)$, one guarantees the same exact bound $\gamma\alpha + \mathrm{L}_{\mathbf{z}}(V, h)$. Moreover, the adversary has no control over $\mathrm{L}_{\mathbf{z}}(V, h)$.

Our model is also robust to what is often called a "white box" adversary that knows and can target $h$ exactly in making its choices. Appendix F.2 has the proof of the above lemma together with an additional analysis showing that if one jointly optimizes $(h, A)$ one achieves a guarantee of $\gamma\alpha + \mathrm{L}_{\mathbf{z}}(V, f)$ where $\mathrm{L}_{\mathbf{z}}(V, f)$ can yield an error bound that is better than that of $\mathrm{L}_{\mathbf{z}}(V, h)$ by a constant factor. In either case, for classification, once nature has picked the natural train and test distributions, $\bar{\mathbf{x}}, \mathbf{z} \sim P^n$, with high probability no matter what fraction $\gamma$ of test examples the adversary corrupts, the learner guarantees loss $\leq \gamma\alpha + \tilde{O}(d/n)$. For regression, the generalization bounds are different of course.

## 2.2 Learning with covariate shift

In the covariate shift setting, there are unknown arbitrary distributions $P, Q$ over train and test examples, respectively. For the reasons mentioned above, we focus on the (transductive) abstention problem where $h$ and version space $V$ are determined from training data, and we focus on the expected test error. Below we give a bound of the form $\alpha|P - Q|_{\mathrm{TV}} + \mathbb{E}_{\mathbf{z} \sim P^n}[\mathrm{L}_{\mathbf{z}}(V, h)]$. Note that the second term is a standard generalization bound for transductive learning with test data from $P$ – the worst case test loss (e.g., $\tilde{O}(d/n)$ for binary classification). Hence, the first term represents the overhead for $P \neq Q$. In fact, we get can state a tighter bound.

We state bounds in terms of $\mathrm{D}_k$ divergence for $k \geq 1$, defined in Eq. (3), which generalizes statistical distance recalling that $\mathrm{D}_1(P\|Q) = |P - Q|_{\mathrm{TV}}$. To see why total variation bounds are loose, consider a distribution $P$ which is uniform on $X = [0, 1]$ and $Q$ which is uniform on $[0, 1/2]$, so $|P - Q|_{\mathrm{TV}} = 1/2$. Clearly the error rate of any $h$ under $Q$ is at most twice the error rate under $P$, and depending on $\alpha$ it may be significantly cheaper not to abstain. In particular, let $\epsilon = \mathbb{E}_{\mathbf{z} \sim P^n}[\ell_{\mathbf{z}}(f, h)]$ be the error of $h$ over $P$. As long as $\epsilon < \alpha/2$, the error upper bound $2\epsilon$ from not abstaining at all is lower than $\alpha|P - Q|_{\mathrm{TV}} = \alpha/2$. The guarantee we give is:

**Lemma 2.** *[Covariate shift] For any distributions* $P, Q$ *over* $X$, *any* $h : X \to Y$, $n \in \mathbb{N}, V \subseteq F$, $f \in V$,

$$\mathbb{E}_{\mathbf{x} \sim Q^n} [\ell_{\mathbf{x}}(f, h, A^*)] \leq \min_{k \geq 1} \alpha \, \mathrm{D}_k(P\|Q) + k \mathop{\mathbb{E}}_{\mathbf{z} \sim P^n} [\mathrm{L}_{\mathbf{z}}(V, h)] \leq \alpha|P - Q|_{\mathrm{TV}} + \mathop{\mathbb{E}}_{\mathbf{z} \sim P^n} [\mathrm{L}_{\mathbf{z}}(V, h)],$$

*where the above holds simultaneously for all* $A^* \in \arg\min_{A \subseteq [n]} \mathrm{L}_{\mathbf{x}}(V, h, A)$.

In the above example, $\mathrm{D}_2(P\|Q) = 0$ for $P, Q$ uniform over $[0, 1]$ and $[0, 1/2]$, respectively.

## 3 The reduction (for classification and regression)

The approach is conceptually simple: it chooses where to abstain so as to minimize the upper-bound on test loss $\mathrm{L}_{\mathbf{x}}(V, h, A)$. The algorithm MMA, in particular, shows that if one can *maximize* loss, then one has a separation oracle that can be used to minimize this upper bound using standard techniques. In particular, the ellipsoid algorithm is used, though simpler algorithms would suffice.[3]

The question then becomes, how difficult is it to find a function in $F$ that *maximizes* loss. For classification, this can be done using an ERM oracle and a label-flipping "trick" used by [10]. For linear regression, this can be done in polynomial time (no ERM oracle is required, and ERM is trivial for linear regression) using existing CDT solvers. We first relax from binary to fractional abstention.

---

[3]Perceptron-like algorithms would suffice as we require only optimization to polynomial (not exp.) accuracy.

---
**Algorithm** MMA
---
Input: $\bar{\mathbf{x}}, \mathbf{x} \in X^n, \bar{\mathbf{y}} \in Y^n, h : X \to Y$, and approximate loss-maximizer $\mathcal{O}$
Output: $\mathbf{a} \in [0,1]^n$
---

Find and output point in convex set $K$:

$$K := \left\{ \hat{\mathbf{a}} \in [0,1]^n : \mathrm{L}_{\mathbf{x}}(\mathrm{VS}(\bar{\mathbf{x}}, \bar{\mathbf{y}}), h, \hat{\mathbf{a}}) \leq \min_{\mathbf{a} \in [0,1]^n} \mathrm{L}_{\mathbf{x}}(\mathrm{VS}(\bar{\mathbf{x}}, \bar{\mathbf{y}}), h, \mathbf{a}) + 1/n \right\}$$

by running the Ellipsoid algorithm using the following separation oracle $\mathsf{SEP}(\mathbf{a})$ to $K$:

1. If $\mathbf{a} \notin [0,1]^n$, i.e., $a_i \notin [0,1]$, then output unit vector $\mathbf{v}$ with $v_i = 1$ if $a_i > 1$ or $v_i = -1$ if $a_i < 0$.

2. Otherwise, output separator

$$\mathbf{v} := \big( c - \ell(g(x_1), h(x_1)), \ldots, c - \ell(g(x_n), h(x_n)) \big),$$

where $g := \mathcal{O}(\bar{\mathbf{x}}, \bar{\mathbf{y}}, h, \mathbf{a})$.     // $g \in \mathrm{VS}(\bar{\mathbf{x}}, \bar{\mathbf{y}})$ maximizes $\ell_{\mathbf{x}}(g, h, \mathbf{a})$ to within $1/(3n)$.

---

Figure 2: The reduction MMA for computing abstention probabilities using an approximate loss maximization oracle $\mathcal{O}$ whose output $g \in \mathrm{VS}(\bar{\mathbf{x}}, \bar{\mathbf{y}})$ maximizes $\ell_{\mathbf{x}}(g, h, \mathbf{a})$ to within $1/3n$. Ellipsoid runtime analysis is in Lemma 3.

**Fractional abstention.**   Loss may be slightly lower if one is allowed fractional abstentions, or equivalently to abstain at random. This can be described by a probability $a_i \in [0,1]$ for each test example $x_i$, which indicates that we abstain with probability $a_i$ and classify according to $h$ with probability $1 - a_i$. As described in the introduction, the definitions of $\ell$ and L are extended to fractional abstention vectors $\mathbf{a} \in [0,1]^n$ in the natural way:

$$\ell_{\mathbf{x}}(f, h, \mathbf{a}) := \frac{1}{n} \sum_{i=1}^n a_i c + (1 - a_i)\ell(f(x_i), h(x_i)) \quad \text{and} \quad \mathrm{L}_{\mathbf{x}}(V, h, \mathbf{a}) := \max_{g \in V} \ell_{\mathbf{x}}(g, h, \mathbf{a}).$$

Our task is to optimize abstention given a fixed $h$ and $\mathbf{x}$ and a fixed set of candidate function $g \in V$ (e.g., a version space), assuming one can *maximize* loss, i.e., find $g \in V$ realizing $\max_{g \in V} \ell(g, h, \hat{\mathbf{a}}) = \mathrm{L}_{\mathbf{x}}(V, h, \hat{\mathbf{a}})$.

**Lemma 3** (Reduction). *For any bounded loss $\ell : Y^2 \to [0,1]$, any $V := \mathrm{VS}(\bar{\mathbf{x}}, \bar{\mathbf{y}}) \subseteq Y^X$, any $h : X \to Y$, and any $\mathbf{x} \in X^n$, the Ellipsoid algorithm can be used in MMA to find $\hat{\mathbf{a}}$ such that,*

$$\mathrm{L}_{\mathbf{x}}(V, h, \hat{\mathbf{a}}) \leq \min_{\mathbf{a}} \mathrm{L}_{\mathbf{x}}(V, h, \mathbf{a}) + \frac{1}{n},$$

*in* $\mathrm{poly}(n)$ *time and calls to approximate loss maximization oracle $\mathcal{O}$ provided that its outputs $g = \mathcal{O}(\bar{\mathbf{x}}, \bar{\mathbf{y}}, h, \mathbf{a}) \in V$ all satisfy $\ell_{\mathbf{x}}(g, h, \mathbf{a}) \geq \max_{g' \in V} \ell_{\mathbf{x}}(g', h, \mathbf{a}) - \frac{1}{3n}$.*

This can be done using any standard optimization algorithm, e.g., the ellipsoid algorithm, with $\mathcal{O}$ serving as a separation oracle. Optimizing to within less than $1/n$ is not generally significant for learning purposes. Note that the Ellipsoid method is more commonly defined with a separation-membership oracle that either claims membership in the convex set $K$ or finds a separator. As we describe in Appendix F.4 containing the proof of Lemma 3, for such optimization problems the Ellipsoid can be used with such an oracle by outputting the query with best objective value.

## 4   Binary classification

For the case of binary classification, the version space is defined as follows:

$$\mathrm{VS}(\bar{\mathbf{x}}, \bar{\mathbf{y}}) := \{g \in F : g(x_i) = y_i \text{ for all } i \leq n\}.$$

We can use generalization bounds on $\mathrm{L}_{\mathbf{z}}(\mathrm{VS}(\bar{\mathbf{x}}, \bar{\mathbf{y}}), h)$ to get directly:

**Theorem 5** (Adversarial classification loss). *For $Y = \{0,1\}$, $d = \mathsf{VC}(F)$, $\ell(y, \hat{y}) = |y - \hat{y}|$, any $n \in \mathbb{N}$, $\delta > 0$, the following holds with probability $\geq 1 - \delta$ over $\bar{\mathbf{x}}, \mathbf{z} \sim P^n$: For all*

$h \in \mathrm{VS}(\bar{\mathbf{x}}, f(\bar{\mathbf{x}}))$, $\mathbf{x} \in X^n$, *and all* $A^* \in \underset{A \subseteq [n]}{\arg\min} \, \mathrm{L}_{\mathbf{x}}(\mathrm{VS}(\bar{\mathbf{x}}, f(\bar{\mathbf{x}})), h, A)$,

$$\ell_{\mathbf{x}}(f, h, A^*) \leq \frac{\alpha}{n} \big| \{i : x_i \neq z_i\} \big| + \frac{2d \lg 2n + \lg 1/2\delta}{n}.$$

*Proof.* This follows directly from Lemma 1 and generalization bound Lemma 7 of Appendix D. $\square$

Covariate shift bounds for classification were stated in Theorems 1 and 2. Next, as observed in prior work [10], for binary classification with the 0-1 loss, one can use an ERM oracle which minimizes loss to *maximize* loss as needed above. This algorithm is called FLIP.

**Lemma 4** (FLIP). *For* $Y = \{0, 1\}$ *with* $\ell(y, \hat{y}) = |y - \hat{y}|$ *and any* $h \in F$, $\bar{\mathbf{x}} \in X^n$, $\bar{\mathbf{y}} = h(\bar{\mathbf{x}}) \in Y^n$, *and any* $\mathbf{a} \in [0, 1]^n$,

$$\hat{g} = \mathsf{ERM}\left(4n^2 \text{ copies of } (\bar{x}_i, \bar{y}_i) \text{ and } \lfloor 3n(1 - a_i) \rfloor \text{ copies of } (x_i, 1 - h(x_i)), \text{ for each } i\right)$$

*satisfies* $\hat{g} \in \mathrm{VS}(\bar{\mathbf{x}}, \bar{\mathbf{y}})$ *and* $\ell_{\mathbf{x}}(\hat{g}, h, \mathbf{a}) \geq \mathrm{L}_{\mathbf{x}}(\mathrm{VS}(\bar{\mathbf{x}}, \bar{\mathbf{y}}), h, \mathbf{a}) - 1/3n$.

In other words, one creates an artificial weighted dataset consisting of numerous copies of the training data $(\bar{x}_i, \bar{y}_i)$ and a number of copies proportional to $1 - a_i$ of each *flipped* test example $(x_i, 1 - y_i)$. Then the classifier output by ERM is in the version space with 0 training error, due to the high weight of each $(\bar{x}_i, \bar{y}_i)$, and approximately maximizes $\ell_{\mathbf{x}}(g, h, \mathbf{a})$. If $\mathcal{O}$ can take weighted examples, then a dataset with only $2n$ samples would suffice. The proof appears in Appendix F.5.

## 5 Regression

In this section, we let $Y = [-1, 1]$ and use the squared loss function $\ell : Y^2 \rightarrow \mathbb{R}_+$, $\ell(y, \hat{y}) = (y - \hat{y})^2$. For regression, since labels are noisy, we will not be able to use an exact version space and instead will need to define an $\alpha$-approximate version space for $\alpha \geq 0$: $\mathrm{VS}_\alpha(\mathbf{x}, h) = \{g \in F \mid \ell_{\mathbf{x}}(g, h) \leq \alpha\}$. For classes $F$ that are convex and have bounded "Rademacher complexity", the version space can be bounded (cf. Section G for the details). Unfortunately, the label flipping "trick" used for classification cannot be applied to maximize loss for regression. The loss here is continuous and convex in $\hat{\mathbf{y}}$, not binary, and maximizing a convex function is intractable in general. For linear functions, which have low Rademacher complexity, however, efficiently maximizing the squared error over certain constrained sets is possible, allowing us to obtain efficient algorithms.

In this subsection, we shall assume that $X = \{x | x \in \mathbb{R}^d, \|x\| \leq 1\}$ and $Y = [-1, 1]$. For $R > 0$, let $F_R = \{x \mapsto w \cdot x \mid w \in \mathbb{R}^d, \|w\| \leq R\}$ be the class of linear function from $X \rightarrow Y$ parametrized by vectors with norm bounded by $R$. Let $F := F_1$.

In order to design an efficient algorithm, we will use an oracle to solve the Celis-Dennis-Tapia problem (CDT): $\min\{f(x) \mid g_1(x) \leq 0, g_2(x) \leq 0, x \in \mathbb{R}^d\}$, where $f$ is a quadratic function, $g_i(x) \leq 0$ for $i = 1, 2$ define ellipsoids and at least one of $g_1, g_2$ is strictly convex. For readability, in the lemma below, we will assume that we have oracle access to *exactly* solve this problem. Theorem 1.3 [3] shows that this problem can be solved in time polynomial in the bit complexity and $\log(\varepsilon^{-1})$, to achieve a solution that is within an additive factor $\varepsilon$ of the optimal and each of the constraints may violate feasibility by additive factor of $\varepsilon$. The lemma as stated below uses an exact oracle to the CDT problem. However, this can easily be addressed by *tightening* the constraint set slightly and using an approximate optimal solution. As the running time is polynomial in $\log(1/\epsilon)$, choosing an $\epsilon$ smaller than the bounds obtained by the learning algorithm suffices. The details are provided in Appendix H.

**Lemma 5** (Regression loss maximization). *For* $X, Y, \ell$ *as defined above, consider* $F$, *the class of linear functions from* $X \rightarrow Y$. *Assuming we have an exact oracle for the CDT problem, for any* $h \in F$, $\bar{\mathbf{x}} \in X^n$, $\bar{\mathbf{y}} \in Y^n$, $\mathbf{x} \in X^n$, $\alpha > 0$, *and* $\mathbf{a} \in [0, 1]^n$, *there exists a polynomial time algorithm that outputs* $\hat{g} \in \mathrm{VS}_\alpha(\bar{\mathbf{x}}, h)$ *such that* $\ell_{\mathbf{x}}(\hat{g}, h, \mathbf{a}) = \mathrm{L}_{\mathbf{x}}(\mathrm{VS}_\alpha(\bar{\mathbf{x}}, h), h, \mathbf{a})$.

## 6 Conclusions

We have shown how transductive abstention can lead to algorithms with robustness to arbitrary covariate shift and adversarial test examples. A simple min-max approach is given to minimize the

worst-case loss and is show how to efficiently implement this approach in certain cases such as linear regression. The min-max approach enjoys both optimal guarantees in terms of test loss and essentially no cost for not knowing the train and test distributions $P$ and $Q$ over $X$.

In terms of future work, it would be interesting to generalize the abstention model to online and possibly agnostic settings, or at least understand the limits. GKKM showed a lower bound of $\Omega(\sqrt{\mathrm{OPT}})$ for one agnostic model, though it is not clear what the right agnostic model is. Finally, it would be interesting to see if their are other applications of our min-max approach of formulating a learning problem as an optimization with no unknowns. It enables one to directly achieve upper bounds as good as if one knows the unknowns.

# 7 Limitations and Societal Impact

Our primary contributions in this paper our mathematical; the contributions, limitations and open problems in mathematical terms have already been discussed in the body of the paper.

Practical machine learning methods may be designed inspired by the algorithms in this paper. When used appropriately, we expect the societal impact of these methods to be largely positive. While in many cases, abstention on out of distribution (OOD) examples may be desirable compared to making an incorrect prediction, it is important that this approach not be used as an excuse to not collect representative data. Potential harm may be done to certain groups by having different rates of abstention among different groups, different rates of misclassification, and potential de-anonymization by revealing examples where a classifier abstained to a human.

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
