# A  Summary of notation

| | |
|---:|:---|
| $X, Y$ | sets of possible examples and labels, resp. |
| $F$ | family of functions $f : X \to Y$ |
| $f \in F$ | unknown target function to be learned |
| $\bar{\mathbf{x}} \in X^n, \bar{\mathbf{y}} \in Y^n$ | $n$ training examples and labels ($\bar{\mathbf{y}} = f(\bar{\mathbf{x}})$ for classification) |
| $\mathbf{x} \in X^n$ | $n$ (unlabeled) test examples |
| $A \subseteq [n]$ | subset of test examples to abstain on |
| $P, Q$ | train and test distributions over $X$ |
| $\nu$ | training distribution for regression over $X \times Y$ with marginal $P$ over $X$ and $f(x) := \underset{(\bar{x}, \bar{y}) \sim \nu}{\mathbb{E}} [\bar{y}|\bar{x}] \in F$ |
| $\alpha$ | cost of abstaining instead of predicting |
| $\ell(y, \hat{y})$ | non-negative base loss function $\ell : Y^2 \to \mathbb{R}_+$ |
| $\ell_{\mathbf{x}}(f, h)$ | test loss $\frac{1}{n} \sum_i \ell(f(x_i), h(x_i))$ |
| $\mathrm{L}_{\mathbf{x}}(V, h)$ | worst possible loss $\max_{g \in V} \ell_{\mathbf{x}}(g, h)$ over $g \in V$ |
| $\ell_{\mathbf{x}}(f, h, A)$ | test loss $\frac{c}{n}|A| + \frac{c}{n} \sum_{i \notin A} \ell(f(x_i), h(x_i))$ |
| $\mathrm{L}_{\mathbf{x}}(V, h, A)$ | worst possible loss $\max_{g \in V} \ell_{\mathbf{x}}(g, h, A)$ over $g \in V$ |
| $\ell_{\mathbf{x}}(f, h, \mathbf{a})$ | test loss $\frac{1}{n} \sum_i c\, a_i + (1 - a_i)\ell(f(x_i), h(x_i))$ |
| $\mathrm{L}_{\mathbf{x}}(V, h, \mathbf{a})$ | worst possible loss $\max_{g \in V} \ell_{\mathbf{x}}(g, h, \mathbf{a})$ over $g \in V$ |
| $\mathrm{VS}(\bar{\mathbf{x}}, \bar{\mathbf{y}}) \subseteq F$ | classification version space $\{g \in F : g(\bar{\mathbf{x}}) = \bar{\mathbf{y}}\}$ |
| $\mathrm{VS}_\alpha(\mathbf{x}, h)$ | regression version space $\{g \in F \mid \ell_{\mathbf{x}}(g, h) \le \alpha\}$ (for $\alpha > 0$) |
| $\mathrm{ERM}(\mathbf{x}, \mathbf{y})$ | deterministic oracle to $\arg\min_{h \in F} \sum_i \ell(y_i, h(x_i))$ for arbitrary $\mathbf{x}, \mathbf{y}$ |
| $h = \mathrm{ERM}(\bar{\mathbf{x}}, \bar{\mathbf{y}})$ | predictor learned based on the labeled training data |

# B  From transduction to generalization

One can trivially convert from transductive learning to generalization. In particular, given a transductive learner that classifies sets of $n$ test examples, and given $n - 1$ test examples, one can output an ordinary classifier $h : X \to Y$ such that $h(x)$ is simply the transductive learner's label on $x$ when $x$ is added to the $n - 1$ examples. The expected errors are the same, as long as the examples are permuted randomly before running the transductive learner. In this section, we point out that this works for abstention as well.

For this section, suppose we have a fixed test distribution $Q$ over $X$, fixed functions $f, h : X \to Y$ with loss $\ell : Y \times Y \to \mathbb{R}$, and an abstention algorithm that outputs a subset of test examples indices, i.e., $A(\mathbf{x}) \subseteq [n]$ for each test set $\mathbf{x} \in X^n$. For this section we will ignore any auxiliary inputs it takes which are chosen independently from $\mathbf{x}$, such as $h$, the labeled training examples, and a version space.

For any distribution $Q$ over $X$, $S \subseteq X$, and $f, h : X \to Y$ we define the expected loss of $h, S$ with respect to target $f$ as,
$$\ell_Q(f, h, S) := \alpha Q(S) + \underset{x \sim Q}{\mathbb{E}} [\mathbf{1}[x \notin S]\, \ell(f(x), h(x))],$$

where $Q(S) := \mathbb{P}_{x \sim Q}[x \in S]$.

**Lemma 6.** *For any* $f, h : X \to Y$, *distribution* $Q$ *over* $X$ *any transductive abstainer* $\mathbf{a} : X^n \to [0, 1]^n$,

$$\mathop{\mathbb{E}}_{\mathbf{x} \sim Q^n, i \in [n]} \left[ \ell_Q\big(f, h, \alpha_i\big) \right] = \mathop{\mathbb{E}}_{\mathbf{x} \sim Q^n} \left[ \ell_{\mathbf{x}}(f, h, \mathbf{a}(\mathbf{x})) \right],$$

*where the expectation is over uniformly random* $i$ *and* $\alpha_i(x) := a_i(x_1, \ldots x_{i-1}, x, x_{i+1} \ldots, x_n)$.

*Proof.* The proof follows from linearity of expectation, the fact that $(x_1, \ldots x_{i-1}, x, x_{i+1}, \ldots, x_n) \sim Q^n$. $\qquad \square$

Since $\ell_Q$ is the generalization loss on future examples $x \sim Q$, this means that one can match the expected loss of any transductive abstainer on future examples drawn from $Q$.

## C    Comparison to GKKM's PQ-learning

The "PQ learning" binary classification model of GKKM is different than Chow's model of learning with a fixed cost $\alpha$ of abstention. In this section, we point out how our algorithm can be used to achieve PQ learning bounds (in expectation) that are similar to those of GKKM, and show how to use their PQ learning algorithm to achieve meaningful but suboptimal loss bounds in our setting of a rejection cost. This section follows GKKM and only considers binary classification with $Y = \{0, 1\}$ and $\ell(y, \hat{y}) = |y - \hat{y}|$.

PQ learning, rather than having a single loss with a fixed cost $\alpha$ for abstentions, considers two separate rates which we denote here by $\epsilon_1$ and $\epsilon_2$:

$$\epsilon_1 = \epsilon_1(Q, h, f, A) := \Pr_{x \sim Q}[h(x) \neq f(x) \wedge x \notin A]$$

$$\epsilon_2 = \epsilon_2(P, A) := \Pr_{x \sim P}[x \in A]$$

The first is the misclassification rate on future test examples from $Q$ (that are not abstained on), and the second (which they call false rejection rate) is the fraction of future examples **from** $P$ that are abstained on. Their Theorem 5.2 bounds $\epsilon_1 + \epsilon_2 \leq \tilde{O}(\sqrt{d/n})$ and their Theorem 5.4 shows $\epsilon_1 + \epsilon_2 \geq \Omega(\sqrt{d/n})$ in the worst case. Hence, in their model, there is a necessary additional cost to abstaining over the $\tilde{O}(d/n)$ rates common in noiseless learning of a $F$ of VC dimension $d$.

First note that their bound directly implies a bound on our loss, defined again as:

$$\ell_Q(f, h, A) := c \Pr_{x \sim Q}[x \in A] + \Pr_{x \sim Q}[h(x) \neq f(x) \wedge x \notin A].$$

Their $\epsilon_1 + \epsilon_2 \leq \tilde{O}(\sqrt{d/n})$ bound gives:

$$\ell_Q(f, h, A) \leq c \left(\epsilon_2 + |P - Q|_{\mathrm{TV}}\right) + \epsilon_1 \leq c|P - Q|_{\mathrm{TV}} + \tilde{O}\left(\sqrt{d/n}\right).$$

This is because the probability of abstaining under $Q$ is at most the probability of abstaining under $P$ plus $|P - Q|_{\mathrm{TV}}$. While their Theorem A.5 gives a trade-off between $\epsilon_1$ and $\epsilon_2$, it only holds for $\epsilon_1 \geq \sqrt{d/n}$ which does not improve the above bound. Varying our parameter $\alpha$ is analogous to their trade-off.

Now, we bound $\mathbb{E}[\epsilon_1 + \epsilon_2] \leq \tilde{O}(\sqrt{d/n})$ using our model. To do so, consider the distribution $Q' = (1 - \lambda)P + \lambda Q$ for $\lambda = \sqrt{d/n} < 1/2$ supposing $n > 4d$. Use the $n$ labeled $P$-samples and $n$ unlabeled $Q$-samples to create a labeled training set of $n/2$ $P$-examples and a synthetic test set of $n/2$ unlabeled $Q'$-samples by, for each test example flipping a $\lambda$-biased coin to determine whether it should be from $P$ or $Q$. From the definitions $Q'$ and $\ell_{Q'}$, we also have,

$$\ell_{Q'}(f, h, A) \geq (1 - \lambda)c\epsilon_2(P, A) + \lambda\epsilon_1(Q, h, f, A).$$

Now, let us choose $\alpha$ so $\lambda = (1 - \lambda)c = \sqrt{d/n}$. Rearranging the above gives,

$$\epsilon_1 + \epsilon_2 \leq \frac{1}{\lambda}\ell_{Q'}(f, h, A).$$

Suppose one finds $h, A$ such that,

$$\ell_{Q'}(f, h, A) \leq c|P - Q'|_{\mathrm{TV}} + \gamma, \tag{6}$$

for some $\gamma$. Combining with the fact that $|P - Q'|_{\mathrm{TV}} \leq \lambda$, gives:

$$\ell_{Q'}(f, h, A) \leq c\lambda + \gamma.$$

Putting this together with the chosen values of $\alpha$ from $\lambda = (1 - \lambda)c = \sqrt{d/n}$, gives,

$$\epsilon_1 + \epsilon_2 \leq c + \frac{\gamma}{\lambda} \leq \frac{\sqrt{d/n}}{1 - \sqrt{d/n}} + \frac{\gamma}{\lambda} \leq \tilde{O}(\sqrt{d/n}) + \frac{\gamma}{\lambda}.$$

Our Theorem 2 shows that our algorithm will output $h, A$ satisfying (6, in expectation) for $\gamma = \tilde{O}(d/n)$. This implies $\mathbb{E}[\epsilon_1 + \epsilon_2] \leq \tilde{O}(\sqrt{d/n})$, similar to GKKM. High probability bounds can be achieved by Markov's inequality.

**Intuition.** The heart of the difference between the two models becomes clear in regions where $P$ and $Q$ overlap, e.g., where $P(x) = \sqrt{d/n}Q(x)$. On such examples, $PQ$-learning either abstains and suffers a $\sqrt{d/n}$ false rejection rate or classifies and may suffer $\sqrt{d/n}$ error rate. In Chow's model with a cost for abstention, the learner can selectively abstain without having to distinguish which examples are from $P$ versus $Q$ which is impossible. See the lower-bound of GKKM for further details.

## D   Transductive bounds for binary classification with $Q = P$

In this section, we bound the expected worst-case test error for binary classification, given training and test sets of size $n$. This is a standard step used in proving generalization bounds, but we give the analysis for completeness. Recall that $L$ is defined in Eq. (5).

**Lemma 7.** *For any $F$ of VC dimension $d$, any $f \in F$, and any distribution $P$ over $X$, and any $\delta \in (0, 1]$,*

$$\mathbb{P}_{\mathbf{x}, \mathbf{z} \sim P^n}\left[ \mathrm{L}_{\mathbf{z}}(\mathrm{VS}(\mathbf{x}, f(\mathbf{x})), f) < \frac{d \lg 2n + \lg 1/\delta}{n} \right] \geq 1 - \delta \tag{7}$$

$$\mathbb{P}_{\mathbf{x}, \mathbf{z} \sim P^n}\left[ \max_{h \in \mathrm{VS}(\mathbf{x}, f(\mathbf{x}))} \mathrm{L}_{\mathbf{z}}(\mathrm{VS}(\mathbf{x}, f(\mathbf{x})), h) < \frac{2d \lg 2n + \lg 1/2\delta}{n} \right] \geq 1 - \delta \tag{8}$$

$$\mathbb{E}_{\mathbf{x}, \mathbf{z} \sim P^n}\left[ \max_{h \in \mathrm{VS}(\mathbf{x}, f(\mathbf{x}))} \mathrm{L}_{\mathbf{z}}(\mathrm{VS}(\mathbf{x}, f(\mathbf{x})), h) \right] \leq \frac{2d \lg 2n}{n}. \tag{9}$$

*Proof.* We first establish Eq. (7). By Sauer's Lemma, there are at most $N = (2n)^d$ labelings of the $2n$ examples $(\mathbf{x}, \mathbf{z}) \in X^{2n}$. For any $k \geq 1$, we claim:

$$\mathbb{P}_{\mathbf{x}, \mathbf{z} \sim P^n}\left[ \max_{g \in \mathrm{VS}(\mathbf{x}, f(\mathbf{x}))} \sum_i |g(z_i) - f(z_i)| \geq k \right] \leq N 2^{-k}.$$

To see the above, as is standard, one can imagine permuting these $2n$ examples without changing their joint distribution. For each of the $< N$ labelings with more than $k$ disagreements $g \neq f$, the probability that all of these disagreements are in $\mathbf{z}$ is at most $2^{-k}$. By the union bound, the probability that any $g$ have all disagreements in $\mathbf{z}$ is $\leq N 2^{-k}$, which is at most $\delta$ for $k \geq d \lg 2n + \lg \frac{1}{\delta}$ as in Eq. (8) of the Lemma.

Similarly, for Eq. (8), by the union bound over the $\binom{N}{2}$ pairs of classifiers,

$$\mathbb{P}_{\mathbf{x}, \mathbf{z} \sim P^n}\left[ \max_{g, h \in \mathrm{VS}(\mathbf{x}, f(\mathbf{x}))} \sum_i |g(z_i) - h(z_i)| \geq k \right] \leq \binom{N}{2} 2^{-k}.$$

By the union bound, the probability that any pair have all disagreements in $\mathbf{z}$ is $\leq N^2 2^{-k-1}$, which is at most $\delta$ for $k \geq 2d \lg 2n + \lg \frac{1}{2\delta}$ as in Eq. (8) of the Lemma.

For Eq. (9), note that for non-negative integer random variable $W$, $\mathbb{E}[W] = \sum_{k=1}^{\infty} \mathbb{P}[W \geq i]$, so,

$$\mathbb{E}_{\mathbf{x},\mathbf{z} \sim P^n} \left[ \max_{g,h \in \mathrm{VS}} \sum_i |g(z_i) - h(z_i)| \right] \leq \sum_{k=1}^{\infty} \mathbb{P}_{\mathbf{x},\mathbf{z} \sim P^n} \left[ \max_{g,h \in \mathrm{VS}} \sum_i |g(z_i) - h(z_i)| \geq k \right]$$

$$\leq \sum_{k=1}^{\infty} \min \left( 1, \binom{N}{2} 2^{-k} \right).$$

Letting $K = \left\lfloor \lg \binom{N}{2} \right\rfloor = \lg \binom{N}{2} - \gamma$ for $\gamma = \lg \binom{N}{2} \in [0,1)$, the last quantity above is,

$$K + \sum_{k=K+1}^{\infty} \binom{N}{2} 2^{-k} = K + \binom{N}{2} 2^{-K} = \lg \binom{N}{2} - \gamma + \lg \binom{N}{2} 2^{-\lg \binom{N}{2} + \gamma} = \lg \binom{N}{2} - \gamma + 2^\gamma.$$

Using the fact that $2^\gamma - \gamma \leq 1$ for $\gamma \in [0,1]$, this gives a bound of at most $1 + \lg \binom{N}{2} = \lg 2\binom{N}{2} \leq \lg N^2$ which implies Eq. (9) of the Lemma since $N = (2n)^d$.

$\square$

# E    High-probability lower bound

In some scenarios, one's goal is minimize expected cost, such as when one regularly trains classifiers and costs are additive. In other prediction scenarios, it is desirable to have high probability guarantees on the accuracy of one's classifier. As noted, Theorem 1 states bounds on *expected* loss rather than more standard high probability bounds. Unfortunately, the following lemma shows that this is for good reason.

**Lemma 8.** *Let $X = \{1, 2\}$, and $F$ be consist of all 4 binary functions on $X$. Fix $P(1) = 1$ and $Q(1) = 1/2$, so $P$ is concentrated on $1$ and $Q$ is uniform. Finally let $\mu$ be the uniform distribution over $F$. There is a constant $\kappa > 0$ such that, for any $c \in (0, 1/2), n \geq 4$, for any transductive selective classification algorithm $L$ selecting $(h, \mathbf{a}) := L(\bar{\mathbf{x}}, f(\bar{\mathbf{x}}), \mathbf{x})$:*

$$\mathbb{P}_{f \sim \mu, \bar{\mathbf{x}} \sim P^n, \mathbf{x} \sim Q^n} \left[ \ell_{\mathbf{x}}(f, h, \mathbf{a}) \geq c|P - Q|_{\mathrm{TV}} + \frac{c}{\sqrt{n}} \right] \geq \kappa.$$

*Proof.* Let $m$ be the number of test 2's. Suppose the selective classifier $h$ predict positively on $r$ of these $m$, negatively on $s$ of them, and abstains on the rest ($r, s \geq 0$ can be arbitrary reals such that $r + s \leq m$ if the classifiers make weighted predictions or if fractional abstentions are allowed). Imagine choosing $f(2)$ after choosing $f(1)$ and $\bar{\mathbf{x}}, \mathbf{x}$. Since the training data reveals nothing about $f(2)$, it is conditionally uniform. Thus with probability $1/2$ over $\mu$, the algorithm's total loss will be $c(m - r - s) + \max(r, s)$. However,

$$c(m - r - s) + \max(r, s) \geq c(m - r - s) + \frac{r+s}{2} = cm + (r+s)\left(\frac{1}{2} - c\right) \geq cm,$$

since $c \in (0, 1/2)$. Thus, with probability at least $1/2$, the algorithm's loss will be $\geq cm$ regardless of its predictions and abstentions. It is well-known that with positive constant probability, say $\geq 2\kappa$, since $m$ is distributed like a standard binomial distribution (hence with standard deviation $\sqrt{n}/2$), $m \geq n/2 + \sqrt{n}$ for $n \geq 4$. Combined with the above, gives that with probability $\geq \kappa$, the average loss will be $\geq c/2 + c/\sqrt{n} = c|P - Q|_{\mathrm{TV}} + c/\sqrt{n}$. $\square$

# F    Deferred proofs

We now prove the remaining deferred proofs.

## F.1    Proofs of main theorems from the introduction

The main theorems are straightforward applications of the other theorems and lemmas in the paper.

*Proof of Theorem 1.* By Lemma 10,

$$\mathop{\mathbb{E}}_{\mathbf{x} \sim Q^n} \left[ \min_A \mathrm{L}_{\mathbf{x}}(V, h, A) \right] \leq \alpha |P - Q|_{\mathrm{TV}} + \mathop{\mathbb{E}}_{\mathbf{z} \sim P^n} [L_{\mathbf{z}}(V, h)],$$

and by Eq. (9) of Lemma 7,

$$\mathop{\mathbb{E}}_{\bar{\mathbf{x}}, \mathbf{z} \sim P^n} [L_{\mathbf{z}}(\mathrm{VS}(\bar{\mathbf{x}}, f(\bar{\mathbf{x}})), h)] \leq \frac{2d \lg 2n}{n}.$$

By Lemma 3 and Lemma 4, MMA together with FLIP find a solution within $1/n$ of optimal, and the proof is completed using the fact that,

$$\frac{2d \lg 2n}{n} + \frac{1}{n} \leq \frac{2d \lg 3n}{n}.$$

$\square$

*Proof of Theorem 2.* This follows just as in the above proof, except using the expectation bound from Lemma 7 and using Lemma 2 instead of Lemma 10. By Lemma 6, the expected error of $\alpha_1$ defined in that lemma is the same as the expected transductive loss $\ell_{\mathbf{x}}(f, h, \mathbf{a}(\mathbf{x}))$. (If the algorithm MMA is not symmetric, then one can shuffle the inputs first to make it symmetric.) $\square$

*Proof of Theorem 3.* This follows from Theorem 5 and Lemmas 3 and 4 and again the fact that,

$$\frac{2d \lg 2n + \lg 1/2\delta}{n} + \frac{1}{n} = \frac{2d \lg 2n + \lg 1/\delta}{n}.$$

$\square$

*Proof of Theorem 4.* Let $h = \mathrm{ERM}(\bar{\mathbf{x}}, \bar{\mathbf{y}})$ which is efficiently computable for linear regression. Clearly, the class of linear functions $F_1 = \{x \mapsto w \cdot x : x, w \in B_d(1)\}$ is convex. It is well-known that it has Rademacher complexity $1/\sqrt{n}$. Thus Lemma 12 shows that, with probability $\geq 1 - \delta$, the version space $V = \mathrm{VS}_\epsilon(\bar{\mathbf{x}}, h)$ both contains $f$ and has $\max_{g \in V} \ell_{\mathbf{z}}(g, h) \leq 14\epsilon$, for $\epsilon = O(\sqrt{\frac{1}{n} \log \frac{1}{\delta}})$. Thus, assuming that the events required for us to be able to apply the conclusion of Lemma 10 hold (which happens with probability at least $1 - \delta$), Lemma 10 shows that,

$$\mathop{\mathbb{E}}_{\mathbf{x} \sim Q^n} \left[ \min_A \mathrm{L}_{\mathbf{x}}(V, h, A) \right] \leq \alpha |P - Q|_{\mathrm{TV}} + \mathop{\mathbb{E}}_{\mathbf{z} \sim P^n} [\mathrm{L}_{\mathbf{z}}(V, h)] \leq \alpha |P - Q|_{\mathrm{TV}} + 14\epsilon.$$

And hence, if MMA minimizes the loss upper-bound of $\hat{A}$ to $1/n$, we will have a loss upper-bound of $c|P - Q|_{\mathrm{TV}} + 14\epsilon + 1/n$. By Lemma 5, one can efficiently maximize loss over the version space if one can solve the CDT problem exactly. As discussed, CDT can be solved to within $\epsilon$ accuracy in time $\mathrm{poly}(\log 1/\epsilon)$. This is more than adequate for the approximate reduction required by Section 3 (see also Appendix H for details regarding the approximation of the CDT problem). To finish the proof, one simply observes that as the loss function is bounded, it suffices to set $\delta = 1/n$. $\square$

### F.2 Proofs from Section 2: information-theoretic loss bounds

In addition to proving Lemma 1, we expand on it to consider the possibility of jointly optimizing $(h, A)$ to minimize the worst-case loss. In particular, the Lemma below includes Lemma 1 as its second part.

**Lemma 9.** *[Adversarial loss (expanded)] For any $n \in \mathbb{N}, V \subseteq F, f \in V, \mathbf{z}, \mathbf{x} \in X^n$,*

$$\ell_{\mathbf{x}}(f, h^*, A^*) \leq \frac{\alpha}{n} \big| \{i : x_i \neq z_i\} \big| + \mathrm{L}_{\mathbf{z}}(V, f) \qquad \text{for all } (h^*, A^*) \in \mathop{\arg\min}_{h \in F, A \subseteq [n]} \mathrm{L}_{\mathbf{x}}(V, h, A) \tag{10}$$

$$\ell_{\mathbf{x}}(f, h, A^*) \leq \frac{\alpha}{n} \big| \{i : x_i \neq z_i\} \big| + \mathrm{L}_{\mathbf{z}}(V, h) \quad \text{for all } h : X \to Y, A^* \in \mathop{\arg\min}_{A \subseteq [n]} \mathrm{L}_{\mathbf{x}}(V, h, A). \tag{11}$$

*Proof.* The idea is that, by minimizing $L_{\mathbf{x}}(V, h, A)$, the resulting loss upper-bound is as low as if one knew $f$ and which points were modified $M := \{i : x_i \neq z_i\}$ and abstained on them. Formally, by definition of L, for all $(h^*, A^*) \in \arg\min_{h \in F, A \subseteq [n]} L_{\mathbf{x}}(V, h, A)$,

$$\ell_{\mathbf{x}}(f, h^*, A^*) \leq L_{\mathbf{x}}(V, h^*, A^*) = \min_{h,A} L_{\mathbf{x}}(V, h, A) \leq L_{\mathbf{x}}(V, f, M).$$

Since $\mathbf{x}$ and $\mathbf{z}$ agree outside of $M$,

$$L_{\mathbf{x}}(V, f, M) = \frac{\alpha}{n}|M| + \max_{g \in V} \frac{1}{n} \sum_{i \notin M} \ell(g(z_i), f(z_i)) \leq \frac{\alpha}{n}|M| + \max_{g \in V} \ell_{\mathbf{z}}(g, f) = \frac{\alpha}{n}|M| + L_{\mathbf{z}}(V, f).$$

In the above we have used the fact that loss is non-negative. This establishes Eq. (10). Similarly,

$$\ell_{\mathbf{x}}(f, h, A^*) \leq L_{\mathbf{x}}(V, h, A^*) = \min_A L_{\mathbf{x}}(V, h, A) \leq L_{\mathbf{x}}(V, h, M).$$

and,

$$L_{\mathbf{x}}(V, h, M) = \frac{\alpha}{n}|M| + \max_{g \in V} \frac{1}{n} \sum_{i \notin M} \ell(g(z_i), h(z_i)) \leq \frac{\alpha}{n}|M| + \max_{g \in V} \ell_{\mathbf{z}}(g, h) = \frac{\alpha}{n}|M| + L_{\mathbf{z}}(V, h).$$

This proves Eq. (11). $\qquad\square$

**Selective prediction versus abstention.** The difference between the two bounds is that in the first case, the learner jointly optimizes for $h$ and $A$, which may be called (transductive) *selective prediction*, while in the second case, $h$ is first fit from the training data and $A$ is selected afterwards, which we refer to as (transductive) *abstention*. Abstention is practically appealing in that it is a post-processing step that can be added to any classifier. As we shall see for classification, the bound (10) is not significantly better than the transductive abstention bounds (11).

### F.3   Covariate shift analysis: Proof of Lemma 2

Before proving Lemma 2, we state a simpler lemma.

**Lemma 10.** *[PQ loss] For any distributions $P, Q$ over $X$, any $h : X \to Y$, $n \in \mathbb{N}, V \subseteq F, f \in V$,*

$$\mathop{\mathbb{E}}_{\mathbf{x} \sim Q^n} [\ell_{\mathbf{x}}(f, h, A^*)] \leq \mathop{\mathbb{E}}_{\mathbf{x} \sim Q^n} \left[ \min_A L_{\mathbf{x}}(V, h, A) \right] \leq \alpha |P - Q|_{\mathrm{TV}} + \mathop{\mathbb{E}}_{\mathbf{z} \sim P^n} [L_{\mathbf{z}}(V, h)],$$

*where the above holds simultaneously for all $A^* \in \arg\min_{A \subseteq [n]} L_{\mathbf{x}}(V, h, A)$.*

Note that this lemma follows directly from a tighter bound we prove in Lemma 2, the proof of this lemma serves as a "warm up" for that Lemma 2's proof.

*Proof.* Let $\mathcal{A}(\mathbf{x}) = \arg\min_{A \subseteq [n]} L_{\mathbf{x}}(V, h, A) \subseteq 2^{[n]}$. The term $\mathop{\mathbb{E}}_{\mathbf{x} \sim Q^n} [\ell_{\mathbf{x}}(f, h, A^*)]$ in the lemma is formally,

$$\mathop{\mathbb{E}}_{\mathbf{x} \sim Q^n} \left[ \max_{A^* \in \mathcal{A}(\mathbf{x})} \ell_{\mathbf{x}}(f, h, A^*) \right] \leq \mathop{\mathbb{E}}_{\mathbf{x} \sim Q^n} \left[ \max_{A^* \in \mathcal{A}(\mathbf{x})} L_{\mathbf{x}}(V, h, A^*) \right] = \mathop{\mathbb{E}}_{\mathbf{x} \sim Q^n} \left[ \min_A L_{\mathbf{x}}(V, h, A) \right]. \quad (12)$$

Thus, since we are minimizing over $A$, it suffices to give a (randomized) procedure for selecting $A$ knowing $\mathbf{x}$ and even $P, Q$ that achieves in expectation,

$$\mathop{\mathbb{E}}_{\mathbf{x} \sim Q^n, A} [L_{\mathbf{x}}(V, h, A)] \leq \alpha |P - Q|_{\mathrm{TV}} + \mathop{\mathbb{E}}_{\mathbf{z} \sim P^n} [L_{\mathbf{z}}(V, h)].$$

To see how, note that it is possible to pick $\mathbf{z} \sim P^n$ by taking modifying an expected $|P - Q|_{\mathrm{TV}}$ fraction of the points in $\mathbf{x}$.[4] As in the proof of Lemma 1, letting $M := \{i : x_i \neq z_i\}$ be the set of modified indices,

$$\min_A L_{\mathbf{x}}(V, h, A) \leq L_{\mathbf{x}}(V, h, M) = L_{\mathbf{z}}(V, h, M) \leq \frac{\alpha}{n}|M| + L_{\mathbf{z}}(V, h).$$

The proof is completed by using the fact that $\mathbb{E}[|M|] = n \cdot |P - Q|_{\mathrm{TV}}$. $\qquad\square$

---

[4]Specifically, for each $i$, choose $z_i = x_i$ with probability $\min(1, P(z_i)/Q(z_i))$ and otherwise choose $z_i$ from the "adjustment" distribution $\rho(x) \propto \max(P(x) - Q(x), 0)$.

*Proof of Lemma 2.* We show the inequality for any $k \geq 1$. We claim it suffices to exhibit joint distribution $(\mathbf{x}, \mathbf{z}, A) \sim \rho$ such that the marginal distribution over $\mathbf{z}$ is $P^n$ and over $\mathbf{x}$ is $Q^n$ and such that:

$$\mathop{\mathbb{E}}_{(\mathbf{x}, \mathbf{z}, A) \sim \rho} [\mathrm{L}_{\mathbf{x}}(V, h, A)] \leq \alpha \mathrm{D}_k(P \| Q) + k \mathop{\mathbb{E}}_{(\mathbf{x}, \mathbf{z}, A) \sim \rho} [L_{\mathbf{z}}(V, h)].$$

This is because, by Eq. (12), $\mathop{\mathbb{E}}_{\mathbf{x} \sim Q^n} [\ell_{\mathbf{x}}(f, h, A^*)]$ is at most,

$$\mathop{\mathbb{E}}_{\mathbf{x} \sim Q^n} \left[ \min_A \mathrm{L}_{\mathbf{x}}(V, h, A) \right] \leq \mathop{\mathbb{E}}_{(\mathbf{x}, \mathbf{z}, A) \sim \rho} [\mathrm{L}_{\mathbf{x}}(V, h, A)]$$
$$\leq \alpha \mathrm{D}_k(P \| Q) + k \mathop{\mathbb{E}}_{(\mathbf{x}, \mathbf{z}, A) \sim \rho} [L_{\mathbf{z}}(V, h)]$$
$$= \alpha \mathrm{D}_k(P \| Q) + k \mathop{\mathbb{E}}_{\mathbf{z} \sim Q^n} [L_{\mathbf{z}}(V, h)].$$

We define $\rho$ by defining a procedure for generating $(\mathbf{x}, \mathbf{z}, A) \sim Q^n$. Begin with $\mathbf{x} \sim Q^n$. The procedure abstains independently on each test example with probability $\alpha(x) := \max \left( 1 - k \frac{P(x)}{Q(x)}, 0 \right)$. The probability of abstaining on test examples $x \sim Q$ is thus $\mathrm{D}_k(P \| Q)$:

$$\mathop{\mathbb{E}}_{\mathbf{x} \sim Q^n} \left[ \frac{|A|}{n} \right] = \sum_{x \in X} Q(x) \max \left( 1 - k \frac{P(x)}{Q(x)}, 0 \right) = \sum_{x \in X} \max \left( Q(x) - kP(x), 0 \right) = \mathrm{D}_k(P \| Q).$$

Thus the expected cost due to abstaining is $c\mathrm{D}_k(P \| Q)$ and it suffices to show that,

$$\mathop{\mathbb{E}}_{(\mathbf{x}, \mathbf{z}, A) \sim \rho} \left[ \max_{g \in V} \sum_{i \notin A} \ell(f(x_i), h(x_i)) \right] \leq k \mathop{\mathbb{E}}_{(\mathbf{x}, \mathbf{z}, A) \sim \rho} [L_{\mathbf{z}}(V, h)]. \tag{13}$$

To complete the description of $\rho$, we now explain how to generate $\mathbf{z}$. For each $i \in A$, we will choose $z_i$ independently from distribution $\mu$ which will be specified shortly. For each $i \notin A$, choose to "copy" $z_i = x_i$ with probability $1/k$ and otherwise, with probability $1 - 1/k$, choose $z_i \sim \mu$.

For any $z \in X$, the probability of choosing any $z_i = z$ is the sum of: (a) the probability of copying $z_i$ from $x_i = z$ which is the probability of choosing $x_i = z$ ($Q(z)$) times the probability of not abstaining $(1 - \alpha(z))$ times the probability of copying $(1/k)$; plus (b) the probability of choosing $z_i \sim \mu$ to be $z$ which is the probability of choosing $z$ from $\mu$ ($\mu(z)$) times the probability of *not* copying (this value $\beta$ is not crucial but it is $\beta := 1 - \frac{1 - \mathrm{D}_k(P \| Q)}{k}$ since probability of copying is the product of the probability of not abstaining $1 - \mathrm{D}_k(P \| Q)$ and copying $1/k$). This yields:

$$\forall i \in [n] : \ \mathbb{P}_{(\mathbf{x}, \mathbf{z}, A) \sim \rho} [z_i = z] = Q(z)(1 - \alpha(z)) \frac{1}{k} + \mu(z)\beta.$$

Since the probability of not abstaining is $1 - \alpha(x) = \min(kP(x)/Q(x), 1)$, the above is

$$\forall i \in [n] : \ \mathbb{P}_{(\mathbf{x}, \mathbf{z}, A) \sim \rho} [z_i = z] = \min(P(z), Q(z)/k) + \mu(z)\beta,$$

which can be made to be $P(z)$ by an appropriate choice of $\mu$, in particular $\mu(z) = \frac{1}{\beta} \max(0, P(z) - Q(z)/k)$.

Since the distribution of $z_i$ is independent across $i$, the above reasoning implies that $\rho$'s marginal distribution over $\mathbf{z}$ is $P^n$. Again, define $M := \{i \in [n] : z_i \neq x_i\}$. For any $\mathbf{x}, A$, pick any

$g^*_{\mathbf{x},A} \in \arg\max_V \sum_{i\notin A} \ell(g(x_i), h(x_i))$. Then we have,

$$\mathbb{E}_{(\mathbf{x},\mathbf{z},A)\sim\rho} [L_{\mathbf{z}}(V,h)] = \mathbb{E}_{(\mathbf{x},\mathbf{z},A)\sim\rho} \left[ \max_{g\in V} \sum_i \ell(g(z_i), h(z_i)) \right] \qquad \text{(by definition of L)}$$

$$\geq \mathbb{E}_{(\mathbf{x},\mathbf{z},A)\sim\rho} \left[ \sum_{i\notin M} \ell(g^*_{\mathbf{x},A}(z_i), h(z_i)) \right]$$

$$= \mathbb{E}_{(\mathbf{x},\mathbf{z},A)\sim\rho} \left[ \sum_{i\notin M} \ell(g^*_{\mathbf{x},A}(x_i), h(x_i)) \right] \qquad \text{(because } x_i = z_i \text{ for } i \notin M)$$

$$\geq \mathbb{E}_{(\mathbf{x},\mathbf{z},A)\sim\rho} \left[ \sum_{i\notin A} \frac{1}{k} \ell(g^*_{\mathbf{x},A}(x_i), h(x_i)) \right] \qquad \text{(because } \mathbb{P}\left[ i\notin M \mid i\notin A \right] \geq 1/k)$$

$$= \frac{1}{k} \mathbb{E}_{(\mathbf{x},\mathbf{z},A)\sim\rho} \left[ \max_{g\in V} \sum_{i\notin A} \ell(g(x_i), h(x_i)) \right] \qquad \text{(by definition of } g^*_{\mathbf{x},A})$$

This is exactly what we needed to show for Eq. (13). $\qquad\square$

### F.4 Proof of Lemma 3 from Section 3: Reduction

Lemma 3 analyzes an algorithm with an oracle SEP that does not determine membership. It is more common to use what we refer to as a "separation-membership" oracle for convex set $K \subseteq \mathbb{R}^n$ (though it is often called simply a separation oracle) is an oracle that given $\mathbf{a} \in \mathbb{R}^n$, in unit time, can both identify which points are in $K$ and which are not, and also for $\mathbf{a} \notin K$, can find $\mathbf{v} \in \mathbb{R}^n$ such that $\mathbf{b} \cdot \mathbf{v} < \mathbf{a} \cdot \mathbf{v}$ for all $\mathbf{b} \in K$. We first argue that the Ellipsoid method would succeed using a separation-membership oracle, which requires just showing circumscribed and inscribed balls.

**Lemma 11.** *Let $\ell : Y^2 \to [0,1]$ be a bounded loss. Fix any $V \subseteq Y^X$, $h \in V$, $\mathbf{x} \in X^n$, and $\epsilon > 0$. Let $\mathrm{OPT} := \min_{\mathbf{a}\in[0,1]^n} L_{\mathbf{x}}(V,h,\mathbf{a})$. Then the Ellipsoid algorithm run with a separation-membership oracle to $K(\epsilon) := \{\mathbf{a} \in [0,1]^n : L_{\mathbf{x}}(V,h,\mathbf{a}) \leq \mathrm{OPT} + \epsilon\}$ will output $\mathbf{a} \in K(\epsilon)$ in time $\mathrm{poly}(n\log 1/\epsilon)$.*

*Proof.* In order to bound the runtime of the Ellipsoid algorithm (see, e.g., [12]), we need only to bound the radii of containing and contained balls for $K(\epsilon)$. Clearly $K(\epsilon)$ is contained in the ball of radius $R = \sqrt{n}$ around the origin. We next argue that $K(\epsilon)$ contains a ball of radius $\geq r = \epsilon/2$. To see this, let $\mathbf{a}^* \in \arg\min_{\mathbf{a}\in[0,1]^n} L_{\mathbf{x}}(V,h,\mathbf{a})$. We claim,

$$Z := \left\{ \mathbf{a} \in [0,1]^n : a_i \in [a_i^* - \epsilon, a_i^* + \epsilon] \text{ for all } i \in [n] \right\} \subseteq K(\epsilon).$$

This is because, for $\mathbf{a} \in K(\epsilon)$,

$$L_{\mathbf{x}}(V,h,\mathbf{a}) = \max_{g\in V} \frac{1}{n} \sum_i \alpha \cdot a_i + (1-a_i)\ell(g(x_i), h(x_i))$$

$$\leq \max_{g\in V} \frac{1}{n} \sum_i \alpha \cdot a_i^* + (1-a_i^*)\ell(g(x_i), h(x_i)) + |a_i - a_i^*| \cdot |c - \ell(g(x_i), h(x_i))|$$

$$\leq L_{\mathbf{x}}(V,h,\mathbf{a}) + \epsilon.$$

The last step follows from the fact that $|a_i - a_i^*| \leq \epsilon$ and that $c, \ell(g(x_i) - h(x_i)) \in [0,1]$. Thus $Z \subseteq K(\epsilon)$. Also, it is not difficult to see that $Z$ contains a cube of side $\epsilon$ and thus also ball of radius $r = \epsilon/2$. So $R/r = 2\sqrt{n}/\epsilon$ and the ellipsoid algorithm runs in $\mathrm{poly}(n\log 1/\epsilon)$ time and queries to a separation oracle outputs $\hat{\mathbf{a}} \in K(\epsilon)$. $\qquad\square$

Using this, we are now ready to prove Lemma 3.

*Proof of Lemma 3.* Let $\mathrm{OPT} := \min_{\mathbf{a}\in[0,1]^n} L_{\mathbf{x}}(V,h,\mathbf{a})$ and, for any $\delta \geq 0$,

$$K(\delta) := \{\mathbf{a} \in [0,1]^n : L_{\mathbf{x}}(V,h,\mathbf{a}) \leq \mathrm{OPT} + \delta\}.$$

It suffices is to output $\hat{\mathbf{a}} \in K(1/n)$. To do so, we fix $\epsilon := 1/(3n)$ simulate running the ellipsoid algorithm on $K(\epsilon)$, which Lemma 11 shows would find $\hat{\mathbf{a}} \in K(\epsilon)$ in $\leq T = \text{poly}(n)$ oracle calls and runtime, using an actual separation-membership oracle to $K(\epsilon)$.

A separation-membership oracle both *separates*: for $\mathbf{a} \notin K(\epsilon)$ it finds a vector $\mathbf{v}$ such that $\mathbf{v} \cdot \mathbf{b} < \mathbf{v} \cdot \mathbf{a}$ for all $\mathbf{b} \in K(\epsilon)$, and computes *membership*: identifying whether or not $\mathbf{a} \in K$. We first argue that SEP separates any $\mathbf{a} \notin K(2\epsilon)$. First, if $\mathbf{a} \notin [0,1]^n$, $\mathbf{v}$ trivially separates $\mathbf{a}$ from $[0,1]^n \supseteq K(\epsilon)$. Next, we argue that $\text{SEP}(\mathbf{a})$ separates any $\mathbf{a} \in [0,1]^n \setminus K(2\epsilon)$ from $K(\epsilon)$. To see this, by definition of $\ell_{\mathbf{x}}(g, h, \mathbf{a})$,

$$\forall \mathbf{b} \in [0,1]^n, \quad \mathbf{v} \cdot \frac{\mathbf{a} - \mathbf{b}}{n} = \ell_{\mathbf{x}}(g, h, \mathbf{a}) - \ell_{\mathbf{x}}(g, h, \mathbf{b}).$$

And thus,

$$\forall \mathbf{b} \in K(\epsilon), \quad \mathbf{v} \cdot \frac{\mathbf{a} - \mathbf{b}}{n} \geq \ell_{\mathbf{x}}(g, h, \mathbf{a}) - (\text{OPT} + 1/3n).$$

On the other hand, by assumption on $\mathcal{O}$, for any $\mathbf{a} \notin K(2\epsilon)$,

$$\ell_{\mathbf{x}}(g, h, \mathbf{a}) \geq \text{L}_{\mathbf{x}}(V, h, \mathbf{a}) - 1/3n > (\text{OPT} + 2/3n) - 1/3n = \text{OPT} + 1/3n.$$

Combining these gives $\mathbf{v} \cdot (\mathbf{a} - \mathbf{b}) > 0$ as needed for all $\mathbf{b} \in K(\epsilon)$ and $\mathbf{a} \in [0,1]^n \setminus K(2\epsilon)$.

The above has shown that SEP separates all $\mathbf{a} \notin K(2\epsilon)$ from $K(\epsilon)$, thus it only fails to be a separation-membership oracle to $K(\epsilon)$ for $\mathbf{a} \in K(2\epsilon)$. Such a failure must occur on one of the first $T$ steps if we run the Ellipsoid algorithm using SEP, because we know the Ellipsoid algorithm would otherwise find a point in $K(\epsilon)$. But such a "failure" $\mathbf{a} \in K(2\epsilon)$ is useful to us as it is in $K(1/n)$. In particular, we adapt the Ellipsoid algorithm as follows. We run it for periods $t = 1, 2, \ldots, T$ using SEP pretending SEP is a separation-membership oracle. Let $\mathbf{a}^{(t)}$ be the $t$-th input to SEP. For $a^{(t)} \in [0,1]^n$, denote $g^{(t)} := \mathcal{O}(\bar{\mathbf{x}}, \bar{\mathbf{y}}, h, \mathbf{a}^{(t)})$. At the end, the algorithm returns $a^{(t)}$ with smallest $\ell_{\mathbf{x}}(g^{(t)}, h, \mathbf{a}^{(t)})$. Again using the assumption on $\mathcal{O}$, we have that,

$$\text{L}_{\mathbf{x}}(V, h, \mathbf{a}^{(t)}) - \epsilon \leq \ell_{\mathbf{x}}(g^{(t)}, h, \mathbf{a}^{(t)}) \leq \text{L}_{\mathbf{x}}(V, h, \mathbf{a}^{(t)}).$$

This means that, among the $\mathbf{a}^{(t)}$, the algorithm outputs one within $\epsilon$ of the smallest $\text{L}_{\mathbf{x}}(V, h, \mathbf{a}^{(t)})$. In particular, since one of them has $\text{L}_{\mathbf{x}}(V, h, \mathbf{a}^{(t)}) \leq \text{OPT} + 2\epsilon$, it outputs one with $\text{L}_{\mathbf{x}}(V, h, \mathbf{a}^{(t)}) \leq \text{OPT} + 3\epsilon = \text{OPT} + 1/n$ as required. $\square$

### F.5 The flipping trick: Proof of Lemma 4

Next, we prove Lemma 4 that shows how to use ERM to approximately maximize test loss, an idea due to GKKM.

*Proof of Lemma 4.* Any $g$ minimizing loss on the artificial dataset must be in $\text{VS}(\bar{\mathbf{x}}, \bar{\mathbf{y}})$ because if $g$ erred on any training example, then its weighted loss would be at least $4n^2$, while loss $\leq 3n^2$ is achievable (any $g \in \text{VS}$). Further, for any $g \in \text{VS}(\bar{\mathbf{x}}, \bar{\mathbf{y}})$,

$$\ell_{\mathbf{x}}(g, h, a) - \ell_{\mathbf{x}}(\hat{g}, h, a) = \frac{1}{3n^2} \sum_i 3n(1 - a_i) \left( \ell(g(x_i), h(x_i)) - \ell(\hat{g}(x_i), h(x_i)) \right)$$

$$\leq \frac{1}{3n^2} \sum_i 1 + \lfloor 3n(1 - a_i) \rfloor \left( \ell(g(x_i), h(x_i)) - \ell(\hat{g}(x_i), h(x_i)) \right)$$

$$= \frac{1}{3n} + \frac{1}{3n^2} \sum_i \lfloor 3n(1 - a_i) \rfloor \left( \ell(\hat{g}(x_i), 1 - h(x_i)) - \ell(g(x_i), 1 - h(x_i)) \right).$$

But the term is at most $1/3n$, as required, since the last summation above is the error difference on the artificial dataset, which is non-positive by definition of ERM. $\square$

### F.6 Proofs from Section 5: Regression

*Proof of Lemma 5.* Note that any $h \in F$ can be parametrized by a vector $w \in \mathbb{R}^d$, such that $\|w\| \leq 1$. Let $w_h$ denote such a parameterization of $h$. Thus, the set $\text{VS}_\alpha(\bar{\mathbf{x}}, h)$ can be characterized as functions

represented by $w \in \mathbb{R}^d$ subject to the following two quadratic constraints:

$$\frac{1}{n} \sum_{i=1}^{n} (w \cdot \bar{x}_i - w_h \cdot \bar{x}_i)^2 \le \alpha, \tag{14}$$

$$\sum_{i=1}^{d} w_i^2 \le 1. \tag{15}$$

These are both ellipsoid constraints; the latter is simply constraining $w$ to lie in the unit ball. In particular, we note that the function $w \mapsto \|w\|^2$ is strictly convex. Recall the definition:

$$\mathrm{L}_{\mathbf{x}}(\mathrm{VS}_\alpha(\bar{\mathbf{x}}, h), h, \mathbf{a}) = \frac{c}{n} \sum_i a_i + \max_{g \in \mathrm{VS}_\alpha(\bar{\mathbf{x}}, h)} \frac{1}{n} \sum_{i=1}^{n} (1 - a_i) \cdot \ell(g(x_i), h(x_i)).$$

As $\mathbf{a}$ is fixed and known, the maximizers $g$ of $\mathrm{L}_{\mathbf{x}}(\mathrm{VS}_\alpha(\bar{\mathbf{x}}, h), h, \mathbf{a})$ are obtained by minimizing the following quadratic function with respect to $w$:

$$-\frac{1}{n} \sum_{i=1}^{n} (1 - a_i)(w \cdot x_i - w_h \cdot x_i)^2,$$

subject to the quadratic inequality constraints (14) and (15). Noting that the inequality constraint (15) is represented by a strictly convex quadratic function, this an an instance of the CDT problem, and can be solved by using the oracle.

$\square$

# G  Regression : Version Space

Let $Y = [-1, 1]$ and consider the squared loss function $\ell : Y^2 \to \mathbb{R}_+$, $\ell(y, \hat{y}) = (y - \hat{y})^2$.

For regression, since labels are noisy, we will not be able to use an exact version space and instead will need to define an $\epsilon$-approximate version space for $\epsilon \ge 0$: $\mathrm{VS}_\epsilon(\mathbf{x}, h) = \{g \in F \mid \ell_{\mathbf{x}}(g, h) \le \epsilon\}$. We first discuss how to bound the version space for regression assuming that the family $F$ has low "Rademacher complexity", which is the case for linear functions. We then show how to efficiently *maximize* loss over this version space, for linear functions. Also, we assume that $F$ is *convex*, which means that for any $f, g \in F$, $\epsilon f + (1 - \epsilon)g \in F$ for every $\epsilon \in [0, 1]$.

For a class of functions $G$, where each $g \in G$ is $g : X \to [0, 1]$, for a set $S = \{x_1, \ldots, x_m\} \subseteq X$, the empirical Rademacher complexity of $G$ with respect to $S$ is defined as,

$$\widehat{\mathrm{RAD}}_S(G) = \mathbb{E}_{\sigma_i \sim \{-1, 1\}} \left[ \sup_{g \in G} \frac{1}{n} \sum_{i=1}^{n} \sigma_i g(x_i) \right],$$

where the $\sigma_i \in \{-1, 1\}$ are chosen uniformly and are known as Rademacher random variables. For any $n \in \mathbb{N}$ and for any distribution $P$ over $X$, define

$$\mathrm{RAD}_n(G) = \mathbb{E}_{S \sim P^m}[\widehat{\mathrm{RAD}}_S(G)].$$

**Lemma 12.** *Let $F$ be a class of functions from $X \to Y$ that is convex. Let $\mathrm{RAD}_n(F)$ denote the Rademacher complexity of $F$. Let $\nu$ be any distribution over $X \times Y$ and let $f \in F$ be such that $\mathbb{E}[y|x] = f(x)$. Let $P$ be the marginal distribution of $\nu$ over $X$. Define,*

$$\varepsilon := 8\mathrm{RAD}_n(F) + \sqrt{\frac{2 \log(3/\delta)}{n}}.$$

*For $(\bar{\mathbf{x}}, \bar{\mathbf{y}}) \sim \nu^n$ and $\mathbf{z} \sim P^n$, if $h \in \arg\min_{g \in F} \ell_{\bar{\mathbf{x}}}(\bar{\mathbf{y}}, g(\bar{\mathbf{x}}))$ then with probability at least $1 - \delta$,*

*1. $f \in \mathrm{VS}_\varepsilon(\bar{\mathbf{x}}, h)$*

*2. $\max_{g \in \mathrm{VS}_\varepsilon(\bar{\mathbf{x}}, h)} \ell_{\mathbf{z}}(g, h) \le 14\varepsilon$*

Before we prove Lemma 12, we state the following lemma.

**Lemma 13.** *Let $F \subseteq Y^X$ be convex. Let $(\bar{\mathbf{x}}, \bar{\mathbf{y}})$ be the training data and $h \in \arg\min_{g \in F} \ell(\bar{\mathbf{y}}, g(\bar{\mathbf{x}}))$. Let $\ell : Y \times Y \to \mathbb{R}^+$ be the squared loss function, $\ell(y, \hat{y}) = (y - \hat{y})^2$. Then for any $f \in F$,*

$$\ell_{\bar{\mathbf{x}}}(f, h) \leq \ell(\bar{\mathbf{y}}, f(\bar{\mathbf{x}})) - \ell(\bar{\mathbf{y}}, h(\bar{\mathbf{x}}))$$

*Proof.* It is straightforward to check that,

$$\ell(\bar{\mathbf{y}}, f(\bar{\mathbf{x}})) - \ell(\bar{\mathbf{y}}, h(\bar{\mathbf{x}})) = \ell_{\bar{\mathbf{x}}}(f, h) + \frac{2}{n} \sum_{i=1}^{n} (f(\bar{x}_i) - h(\bar{x}_i))(h(\bar{x}_i) - y_i)$$

We observe that the second term on the RHS above is non-negative, otherwise, $g := (1 - \epsilon)h + \epsilon f$ for a sufficiently small $\epsilon > 0$ would have $\ell(\bar{\mathbf{y}}, g(\bar{\mathbf{x}})) < \ell(\bar{\mathbf{y}}, h(\bar{\mathbf{x}}))$ contradicting the optimality of $h$. $\square$

*Proof of Lemma 12.* Note that $f \in F$ is such that $\mathbb{E}_D[y|x] = f(x)$. First consider the class of functions $F^1 := \{(x, y) \mapsto \frac{1}{4}\ell(g(x), y) \mid g \in F\}$ and $F^2 := \{x \mapsto \frac{1}{4}\ell(g(x), f(x)) \mid g \in F\}$. By using Talagrand's lemma and the fact that $x \mapsto \frac{x^2}{4}$ is $1/2$-Lipschitz for $x \in [0, 1]$, we get that $\mathsf{RAD}_m(F^1) \leq \frac{1}{2}\mathsf{RAD}_m(F)$ and $\mathsf{RAD}_m(F^2) \leq \frac{1}{2}\mathsf{RAD}_m(F)$ (see e.g. Lemma 5.7 from [15]). For a distribution $\nu$ over $X \times Y$, we denote by $\ell_\nu(g) = \mathbb{E}_{(x,y) \sim \nu}[\ell(y, g(x))]$. For the value of $\varepsilon$ set in the statement of the lemma, we have with probability at least $1 - \delta/3$, each of the following hold for every $g \in F$ (cf. Theorem 3.3 from [15]):

$$|\ell_\nu(g) - \ell(\bar{\mathbf{y}}, g(\bar{\mathbf{x}}))| \leq \varepsilon/2, \tag{16}$$
$$|\ell_P(f, g) - \ell_{\bar{\mathbf{x}}}(f, g)| \leq \varepsilon/2, \tag{17}$$
$$|\ell_P(f, g) - \ell_{\mathbf{z}}(f, g)| \leq \varepsilon/2. \tag{18}$$

By a simple union bound, all of the above hold for except with probability at most $\delta$. For the rest of the proof, we assume that the failure event does not occur.

Then, using Lemma 13, we have the following,

$$
\begin{aligned}
\ell_{\bar{\mathbf{x}}}(f, h) &\leq \ell_{\mathbf{x}}(\bar{\mathbf{y}}, f(\bar{\mathbf{x}})) - \ell_{\bar{\mathbf{x}}}(\bar{\mathbf{y}}, h(\bar{\mathbf{x}})) \\
&\leq \ell_\nu(f) - \ell_\nu(h) + \varepsilon && \text{Using (16)} \\
&\leq \varepsilon. && \text{As } \ell_\nu(f) = \min_{g \in F} \ell_\nu(g)
\end{aligned}
$$

This proves the first part of the result. For the second part, we will make repeated use of the following crude inequality when $\ell$ is the squared loss: for any $h \in F$, $\ell_{\mathbf{x}}(f, g) \leq 2\ell_{\mathbf{x}}(f, h) + 2\ell_{\mathbf{x}}(h, g)$. An analogous inequality holds when considering $\ell_P$. For the rest of the proof, denote by $V = \mathsf{VS}_\varepsilon(\bar{\mathbf{x}}, h)$. Then, we have

$$
\begin{aligned}
\max_{g \in V} \ell_{\mathbf{z}}(g, h) &\leq 2\ell_{\mathbf{z}}(f, h) + 2\max_{g \in V} \ell_{\mathbf{z}}(g, f) \\
&\leq 2\ell_P(f, h) + 2\max_{g \in V} \ell_P(g, f) + 2\varepsilon && \text{Using (18)} \\
&\leq 2\ell_{\bar{\mathbf{x}}}(f, h) + 2\max_{g \in V} \ell_{\bar{\mathbf{x}}}(g, f) + 4\varepsilon && \text{Using (17)} \\
&\leq 6\ell_{\bar{\mathbf{x}}}(f, h) + 4\max_{g \in V} \ell_{\bar{\mathbf{x}}}(g, h) + 4\varepsilon && \text{Using } \ell_{\bar{\mathbf{x}}}(g, f) \leq 2\ell_{\bar{\mathbf{x}}}(g, h) + 2\ell_{\bar{\mathbf{x}}}(h, f) \\
&\leq 10\max_{g \in V} \ell_{\bar{\mathbf{x}}}(g, h) + 4\varepsilon \leq 14\varepsilon.
\end{aligned}
$$

In the last line above we used the fact that $f \in V$ to upper bound $\ell_{\bar{\mathbf{x}}}(f, h)$ by $\max_{g \in V} \ell_{\bar{\mathbf{x}}}(g, h)$. $\square$

# H   Approximately solving the CDT Problem

By slight abuse of notation, we will use the letter $g$ (suitably annotated) to denote a linear function and $w$ to denote the weight vector associated with the same linear function without explicit reference, e.g. $\hat{g}$ is associated with $\hat{w}$, $\tilde{g}$ is associated with $\tilde{w}$, etc. Let $\hat{g}$ be the solution obtained by *exactly* solving the CDT problem as stated in Lemma 5.

We can instead tighten the constraints (14) and (15) by requiring that $w$ satisfy,

$$\frac{1}{n}\sum_{i=1}^{n}(w\cdot\bar{x}_i - w_h\cdot\bar{x}_i)^2 \leq \alpha - \epsilon, \tag{19}$$

$$\sum_{i=1}^{d}w_i^2 \leq 1-\epsilon. \tag{20}$$

Recall that the function we are minimizing in the CDT problem is given by:

$$F(w) = -\frac{1}{n}\sum_{i=1}^{n}(1-a_i)(w\cdot x_i - w_h\cdot x_i)^2. \tag{21}$$

Let $\tilde{g}$ denote a solution to the CDT problem with constraints (19) and (20) obtained using an approximate solver, which may violate the constraints by an additive factor of $\epsilon$. In particular, this means that $\tilde{g}$ satisfies the constraints (14) and (15). Furthermore, if $\hat{g}_\epsilon$ is the solution obtained by the exactly solving the CDT problem with constraints (19) and (20), then we know that $F(\tilde{w}) \leq F(\hat{w}_\epsilon) + \epsilon$, again by the approximation guarantees given by the result of [3].

Let $\hat{g}_{\lambda,\delta} = (1-\delta)((1-\lambda)\hat{g} + \lambda h)$. It can easily by checked that if we set $\lambda = 9\epsilon/\alpha$ and $\delta = \epsilon$, then using the fact that $F(w)$ is 8-Lipschitz for $w$ in the unit ball, $\hat{g}_{\lambda,\delta}$ satisfies the constraints (19) and (20). However, we note that $\|\hat{w}_{\lambda,\delta} - \hat{w}\|_2 = O(\lambda + \delta)$. Thus, we have, again using the Lipschtizness of $F$,

$$\begin{aligned}F(\tilde{g}) &\leq F(\hat{g}_\epsilon) + \epsilon \\ &\leq F(\hat{g}_{\lambda,\delta}) + \epsilon \\ &\leq F(\hat{g}) + O(\lambda + \delta).\end{aligned}$$

The only observation that remains to be made is that $\epsilon$ may be set to be as small as we please since the running time of the approximate solver to the CDT problem provided by [3] runs in time polynomial in $\log(1/\epsilon)$. Certainly, this is more than sufficient for the results that we need to apply from Section 3.