# OpenReview forum: "Towards optimally abstaining from prediction with OOD test examples"
_NeurIPS.cc/2021/Conference — NeurIPS 2021 Spotlight_

### Official Review · Reviewer_UTdk · 2021-07-09

**Rating:** 7
**Confidence:** 3

**Summary:**

The paper studies selective prediction where abstention incurs a fixed and known cost. It extends the Fundamental Theorem of Statistical Learning to selective classification and linear regression under distributional shift. The paper also proposes min-max abstention reduction with optimal guarantees in test loss.

**Limitations And Societal Impact:**

Without concrete examples, it is hard to understand why the proposed algorithm and all the theorems are important. Besides, it may be better to describe in the beginning what the problem/challenge is that the paper tries to tackle, instead of directly diving into details. Empirical evaluation to demonstrate the importance of the theoretic results and to show the effectiveness of the proposed algorithm will also be highly helpful.

**Main Review:**

The paper builds upon GKKM to study selective classification and regression, with adversarial test examples and covariate shift.
It provides not only bounds for test loss, but also an efficient algorithm for abstaining in linear regression and an efficient reduction for binary classification. Although there is little empirical verification, the results provide decent theoretical value.

**Time Spent Reviewing:**

4

---

> ### Author Response · Authors · 2021-08-05
> **Review Response**
>
> We thank you for the helpful review. We agree that a clear example would better motivate our work. We will better flesh out the medical example in the second paragraph: If an “abstention” in a diagnostic test results in a further referral or a better test, then the cost of an error may be much greater than the cost of abstention (possibly time and more expensive testing)---errors may in fact be (unnecessarily) life-threatening. The setting of covariate shift is also natural in this context, as it is unlikely that the data used to design the test will be exactly representative of the population on which the tests are used.
>
> Depending on space considerations, we will add other examples, but at least refer to the section in GKMM where related examples are discussed.

---

### Official Review · Reviewer_Lekf · 2021-07-16

**Rating:** 6
**Confidence:** 3

**Summary:**

This paper studies agnostic transductive learning when train and test distributions have covariate shift. The objective follows Chow 57', where abstention suffers a fixed cost, and the goal is to minimize total costs of prediction error and abstentions. The paper proposes efficient algorithms for linear regression and binary classification, based on ideas from Goldwasser et al. The paper also proves lower bounds showing that the guarantee is near-optimal.

**Main Review:**

The paper is well-written, with clear motivations and comparison with previous works. The theoretical parts are rigorous, and the proofs are modularized and easy to follow. There are a few weaknesses:

1. The algorithms only work for linear models, and do not seem to be able to generalize to more expressive models.
2. Techniques for binary classification are mostly from Goldwasser et al., although the ellipsoidal algorithm for regression is new.
3. The algorithms rely on poly(n) calls to an ERM procedure on poly(n) examples, which may be time-consuming in practice.

Typos:
1. Line 323.


**Time Spent Reviewing:**

3

---

> ### Author Response · Authors · 2021-08-05
> **Review response**
>
> Thank you for the helpful review. We’d like to point out that both our classification and regression algorithms in fact do work for all models, not just linear models. However, the algorithms in general will depend on oracles that may or may not be efficiently implementable. As it stands, the algorithms we present are efficient only for linear regression (through the CDT problem) and for binary classification whenever ERM is tractable.
>
> We also mention that the use of ellipsoid is new both for classification and regression. For classification, Goldwasser et al. consider a greedy set-cover-like algorithm, called Rejectron, to decide which examples to abstain on. We replace their approximation-algorithm approach with one based on exact min-max optimization.

---

> > ### Comment · Reviewer_Lekf · 2021-08-30
> > **Thanks for the response**
> >
> > Thanks for your clarification. I will keep my original score.

---

### Official Review · Reviewer_zQKS · 2021-07-17

**Rating:** 6
**Confidence:** 4

**Summary:**

The paper considers prediction with the abstention option, with a reduced loss for abstention (Chow's model), in a setting where the test distribution can be different from the training distribution. The setting is very similar to GKKM20 except that the loss here is different (GKKM separately bounds standard risk and amount of abstention, current paper minimizes a weighted combination), and current work considers linear regression in addition to binary classification. The paper presents efficient algorithms given ERM oracles and bounds the Chow's loss of their procedures for both binary classification and linear regression. The technique involves setting abstention to minimize the abstention-equipped loss, by using a separation oracle that needs an approximate maximizer of the loss. The upper bounds depend on the variational distance between the test and training distributions plus a term which decreases with sample size.

**Limitations And Societal Impact:**

I did not find a discussion of limitations, potential societal impact is reasonably addressed.

**Main Review:**

The paper uses a general reduction algorithm for computing the min-max abstention. The paper minimizes Chow's loss with an efficient reduction to loss maximization using a separation oracle. Learning with covariate shift and adversarial examples is a significant research question, and modeling with reduced cost of abstention may be useful in some cases. The comparison to prior work GKKM20 and a summary of the results is clearly stated.

Limitations: The use of Chow's risk model gives faster convergence than GKKM20 which the current paper references throughout. However similar fast convergence is also known in prior work using Chow's risk in standard learning (without covariate shift) in [1] below which the authors do not cite. It would be useful to understand how the techniques differ. Also it is unclear to me in what sense are the bounds optimal, since there do not seem to be any lower bounds. The min-max approach is conceptually straightforward and use prior work as oracles. For the binary classification the reduction is polynomial time, but still depends on an ERM.

[1] O. Bousquet, N. Zhivotovskiy. Fast classification rates without standard margin assumptions, 2021. Information and Inference: A Journal of the IMA

-----------------------------
Post rebuttal edit:

I thank the authors for their response and have increased my rating. While it is still a concern for me that an important related work was missed by the authors, the additional clarifications and changes proposed by the authors are useful. I also suggest adding a paragraph or section discussing limitations of the work (as suggested in NeurIPS guidelines).

**Time Spent Reviewing:**

4

---

> ### Author Response · Authors · 2021-08-05
> **Review response**
>
> Thank you for the helpful review. We agree and will emphasize that our paper is focused on optimality in the case of out-of-distribution (OOD*) covariate shift, and we will add "with OOD test examples" to the title to make it "Towards optimally abstaining from prediction with OOD test examples." We will add a reference to [1] which studies chow loss without covariate shift. We will also mention the sense in which our bounds are optimal -- for the case of transductive abstention, the approach by definition exactly minimizes the worst-case transductive loss among (fractional) abstention algorithms. For other settings, such as the one studied in [1] and others, it may not be optimal. Thank you for pointing out [1] as some of the techniques there may be useful to further improve our analysis. A key point of difference is that the aggregation technique in [1] by itself is not sufficient for our problem: because we are dealing with OOD test examples we do need to frame it as a min-max problem.
>
> *OOD is a standard term, used by reviewer 5Vsa, that we missed in our writeup.

---

### Official Review · Reviewer_5Vsa · 2021-07-19

**Rating:** 8
**Confidence:** 4

**Summary:**

This paper considers the abstention model in which a classifier is able to abstain from predicting (and thus paying a fixed cost) as opposed to being forced to give a prediction.

They consider a formalism in which the algorithm is given a classifer $f$ with 0 training loss, and it must make predictions for an additional $n$ test points. The classifier incurs a loss of 1 for each error it makes on the test points, but incurs loss $\alpha$ if it chooses to abstain (where $\alpha \leq 1$). Furthermore, the points in the test set are drawn from some distribution that differs from the original training distribution.

In this formalism, they give a generalization bound of a polynomial time algorithm that uses empirical risk minimization as a subroutine. The basic gist is that the overall expected lost can be bounded by the total variation between the training and test distributions along with a $\frac{O(d\logn}{n}$ term where $d$ is the VC-dimension of the classifiers.

They then apply their ideas to the setting of adversarial classification, in which adversarial examples can be thought of as being drawn from a distribution $Q$ that is very close to the original data disribution $P$. They also apply their theorem to linear regression giving a bound on how well a linear classifier can be learned.

**Limitations And Societal Impact:**

Yes.

**Main Review:**

This paper provides an elegant formalism along with a relatively decent performing algorithm. It is a non-trivial result that also leaves a lot of room for further work. I also find this abstraction particularly relevant to the open problem of defending against adversarial examples, which can be thought of as a form of OOD generalization.

There are a few smaller ideas in the paper I liked a lot as well. For example, if the test risk was defined with respect to a single o.o.d point, it would be difficult to make much progress as a single point could plausibly be drawn from any distribution. By having the algorithm simultaneously label $n$ points, it is able to adopt deeper strategies about when it chooses to abstain, and when it chooses to label. In particular, it feels important that the algorithm is able to learn from a new test distribution by just seeing unlabeled examples.

**Time Spent Reviewing:**

2 hours

---

> ### Author Response · Authors · 2021-08-05
> **Review response**
>
> Thank you for the insightful review. We appreciate your summary that defending against adversarial examples can be thought of as a form of OOD generalization. In fact, the common "OOD" term is helpful and we will use it in the paper to better explain our setting.

---

### Decision · Program_Chairs · 2021-09-27

**Decision:**

Accept (Spotlight)

**Comment:**

This paper considers a setting in which one may have different train/test distributions, but the predictor can choose to abstain at a cost of \alpha. All reviewers believed that the results were interesting, non-trivial, and clearly-stated, and all four recommended acceptance. What criticisms there were were adequately addressed in the author response.

Please be sure to make any changes that have been promised to the reviewers, and seriously consider attempting to address any other concerns that they might have raised, in particular  UTdk's request for concrete examples.